# Investigation of ice cloud modelling capabilities for the irregularly shaped Voronoi ice scattering models in climate simulations

Ming Li[1], Husi Letu[1*], Yiran Peng[3], Hiroshi Ishimoto[4], Yanluan Lin[3], Takashi Y. Nakajima[2], Anthony Baran[5,6], Zengyuan Guo[3,7], Yonghui Lei[2], Jiancheng Shi[8]

[1]Aerospace Information Research Institute, Chinese Academy of Sciences, Beijing 100010, China
[2]Research and Information Center (TRIC), Tokai University, 4-1-1 Kitakaname Hiratsuka, Kanagawa 259-1292, Japan
[3]Ministry of Education Key Laboratory for Earth System Modeling, Department of Earth System Science, Tsinghua University, Beijing 10084, China
[4]Meteorological Research Institute, Japan Meteorological Agency (JMA), Nagamine 1-1, Tsukuba 305-0052, Japan
[5]Met Office, Fitzroy Road, Exeter EX1 3PB, UK
[6]School of Physics, Astronomy and Mathematics, University of Hertfordshire, Hatfield, AL10 9AB, UK
[7]Laboratory for Climate Studies, National Climate Center, China Meteorological Administration, Beijing 100081, China
[8]National Space Science Center, Chinese Academy of Sciences, Beijing 100190, China

*Correspondence to*: Husi Letu (husiletuw@hotmail.com)

**Abstract.**

Both weather/climate models and ice cloud remote sensing applications all need to obtain effective ice crystal scattering (ICS) properties and the parameterization scheme. An irregularly shaped Voronoi ICS model has been suggested to be effective in remote sensing applications for several satellite programs, e.g., Himawari-8, GCOM-C (Global Change Observation Mission-Climate) and EarthCARE (Earth Cloud Aerosol and Radiation Explorer). As continuation work of Letu et al. (2016), an ice cloud optical property parameterization scheme (Voronoi scheme) of the Voronoi ICS model is employed in the Community Integrated Earth System Model (CIESM) to simulate the optical and radiative properties of ice clouds. We utilized the single-scattering properties (extinction efficiency, single-scattering albedo and asymmetry factor) of the Voronoi model from the ultraviolet to the infrared, combined with 14,408 particle size distributions obtained from aircraft measurements to complete the Voronoi scheme. The Voronoi scheme and existing schemes (Fu, Mitchell, Yi and Baum-yang05) are applied to the CIESM to simulate 10-yr global cloud radiative effects during 2001-2010. Simulated global-averaged cloud radiative forcing at the top of the atmosphere (TOA) for Voronoi and other four existing schemes are compared to the Clouds and the Earth's Radiant Energy System Energy Balanced And Filled (EBAF) product. The results show that the difference in shortwave and longwave global-averaged cloud radiative forcing at the TOA between the Voronoi scheme simulations and EBAF products is 1.1% and 1.4%, which is lower than that of the other four schemes.

Particularly for regions (from 30°S to 30°N) where ice clouds occur frequently, the Voronoi scheme provides the closest match with EBAF products compared with other four existing schemes. The results in this study fully demonstrated the effectiveness of the Voronoi ICS model in the simulation of the radiative properties of ice clouds in the climate model.

## 1 Introduction

Ice clouds cover about 20% - 30% of the global area (Rossow and Schiffer, 1991; Wang et al., 1996; Stubenrauch et al.,
2013), and they strongly affect the earth's energy budget and climate system mainly due to their optical and radiative properties (Liou, 1986, 1992; Baran, 2012; Ramaswamy and Ramanathan, 1989). The radiative properties of ice clouds mainly depend on their optical properties (e.g., scattering albedo and optical thickness), which are significantly influenced by the microphysical properties (e.g., ice particle sizes and habits) of ice clouds (Baran, 2009; Yang et al., 2015; Yang et al., 2018). Based on the accurate knowledge of ice particle sizes and habits, the single-scattering properties (e.g., extinction
efficiency, single-scattering albedo and asymmetry factor) of ice particles can be calculated by using the light scattering computational method to develop the ice crystal scattering (ICS) model/database. Combined with the single-scattering properties of the ICS model/database and size distributions, the optical properties of ice clouds can be simulated through the parameterization scheme, and the radiative properties of ice clouds can be further simulated based on the radiative transfer theory.
At present, satellite remote sensing and weather/climate models are two effective ways to understand the ice cloud optical and radiative properties through the ice cloud optical property parameterization scheme in radiative transfer models. However, numerous field observations (Rossow and Schiffer, 1999; Lawson et al., 2006; Heymsfield et al., 2017; Lawson et al., 2019), e.g., the First International Satellite Cloud Climatology Project (ISCCP) Regional Experiments (FIRE-I) in 1986 and 1991 (Rossow and Schiffer, 1999) and the European cirrus experiment in 1989 (Liou, 1992) have shown that ice clouds
contain a large variety of non-spherical ice particle sizes and habits, which can lead to inaccurate simulations of the optical and radiative properties of ice clouds in nature. Our understanding of how ice particle habits affect the optical and radiative properties of ice clouds is still limited (Heymsfield and Miloshevich, 2003; van Diedenhoven et al., 2014b; van Diedenhoven, 2018). The current insufficient knowledge of the ice cloud microphysical properties and the ice particle single-scattering

properties contributes to inadequate representation of the optical properties in the parameterization scheme, which can directly lead to uncertainties in the simulated radiative properties of ice clouds (Zhang et al., 2015; Yang et al., 2015, 2018; Yi et al., 2017; van Diedenhoven and Cairns, 2020). Thus, the accurate representation of the microphysical properties of ice clouds and ice particle single-scattering properties is essential for the parameterization of ice cloud optical properties and studying the radiative properties of ice clouds in both satellite remote sensing and weather/climate models.

In terms of current light scattering computational methods, it is still difficult for one specific method to accurately calculate the single-scattering properties for non-spherical particles with different size parameter (SZP), which is defined as the ratio of the equivalent-volume sphere's circumference dimension (or $\pi$ times particle maximum diameters) to the incident wavelength (Nakajima et al., 2009; Baran, 2012; Yang et al., 2015). The existing light scattering computational methods for non-spherical particles can be generally divided into the approximation method (AM) based on the ray-tracing techniques (Wendling et al., 1979), and the numerical simulation (NM) method based on the approximate solutions of Maxwell equations. The AM method is suitable for non-spherical particles with very large SZPs. The geometrical optics approximation (GOA) method is a typical AM method. This method can capture the halo phenomenon of large hexagonal ice particles in the visible wavelength. However, the AM method is difficult to accurately simulate the single-scattering properties for particles with small and moderate SZPs. The NM method is suitable for particles with small SZPs and can be divided into the volume and surface-based methods depending on how Maxwell equations are solved. The volume-based NM method includes the finite-difference time domain (FDTD) (Yee, 1966; Yang and Liou, 1996b) and discrete dipole-approximation (DDA) methods (Draine and Flatau, 1994; Yurkin and Hoekstra, 2007). A typical method of the surface-based method is the T-matrix method (Havemann and Baran, 2001; Mishchenko and Travis, 1998). However, the NM method requires discretization of the whole volume/surface of the scatterer and needs rather high computational demands (Nakajima et al., 2009), so it is difficult to efficiently calculate the single-scattering properties for particles with moderate and large SZPs. Later, the invariant imbedding T-matrix (II-TM) (Bi et al., 2013a; Bi and Yang, 2014) and physical-geometric optics hybrid (PGOH) method (Bi et al., 2011) are developed for particles with small to moderate SZPs. Combined with the advantages of the AM and NM methods, several improved GOA methods including the geometric optics integral equation (GOIE) (Yang and Liou, 1996; Ishimoto et al., 2012a) and improved geometric-optics method (IGOM)

(Yang and Liou, 1995, 1996a; Bi et al., 2010) have been developed. The GOIE and IGOM methods are useful for particles with moderate SZPs, and therefore they can bridge the gap between the AM and NM methods. Thus, the light scattering computation of particles with different SZPs can be completed by a combination of the AM and NM methods.

With the development of the ICS model/database, numerous parameterization schemes of the ice cloud optical properties have been developed for use in the ice cloud remote sensing and weather/climate model applications (Yang et al., 2015, 2018). In terms of weather/climate model applications, Fu (1996) developed a parameterization scheme (referred to as the Fu scheme hereafter) using the GOA-based ICS database for the randomly oriented hexagonal particle (Takano and Liou, 1989). The Fu scheme was subsequently applied to the Fu-Liou radiative transfer model for use in the climate models (Fu, 1996, 2007). Mitchell et al. (1996b, 2006) used the modified anomalous diffraction approximation (MADA) method (Mitchell and Arnott, 1994) to generate an ICS database for a habit mixture and completed a parameterization scheme (referred to as the Mitchell scheme hereafter) combined with the bimodal size distributions (Mitchell et al., 1996a). The Mitchell scheme was then employed in the National Center for Atmospheric Research Community Atmosphere Model (CAM). Yang et al. (2000a) used the IGOM and FDTD methods to develop an ICS database for six ice particle habits. However, this database contains several inconsistencies in the spectral regions caused by differences of particle habits and computational methods. Later, Yang et al. (2013) utilized the Amsterdam DDA (Yurkin et al., 2007; Yurkin and Hoekstra, 2011), T-matrix (Mishchenko et al., 1996) and improved IGOM (Bi et al., 2009) methods to generate a spectrally consistent ICS database for 11 ice particle habits. Yi et al. (2013) employed the ICS database of Yang et al. (2013) and developed a parameterization scheme (referred to as the Yi scheme hereafter) for use in the CAM. For ice cloud remote sensing applications, Baum et al. (2005a, 2005b) used the ICS database of Yang et al. (2000a) to develop a parameterization scheme (referred to as the Baum-yang05 scheme hereafter) for the MODIS collection 5 ice cloud product. C.-Labonnote et al. (2000, 2001) and Doutriaux-Boucher et al. (2000) developed an ICS database for the Inhomogeneous Hexagonal Monocrystal (IHM) containing embedded inclusions (air bubbles and aerosols) and developed a parameterization scheme for use in the ice cloud retrievals from the French satellite Polarization and Directionality of the Earth's Reflectance (POLDER) measurements (Deschamps et al., 1994). Ishimoto et al., (2012) and Letu et al. (2016) developed an ICS database by using a combination method of the FDTD, GOIE and GOM for an irregularly shaped Voronoi model based on in situ microphysical

measurements. Letu et al. (2016, 2020) demonstrated that the Voronoi model can effectively retrieve the ice cloud microphysical properties from satellite measurements. Furthermore, the Voronoi model has been adopted for generating official ice cloud products for the Second Generation gLobal Imager (SGLI)/Global Change Observation Mission-Climate (GCOM-C) (Letu et al., 2012, 2016; Nakajima et al., 2019), AHI/Himawari-8 (Letu et al., 2018) and the Multi-Spectral Imager (MSI)/Earth Cloud Aerosol and Radiation Explorer (EarthCARE) satellite programs (Illingworth et al., 2015), which will be launched in 2023. These studies demonstrated the superiority of the Voronoi model in the ice cloud remote sensing applications. However, the performance of the Voronoi model in climate model simulations has not been investigated quantitatively.

Motivated by the abovementioned situations, this study aims to quantify the effects of the Voronoi model on the optical and radiative properties of ice clouds in a climate model in comparison with other ice cloud optical property schemes (Fu, Mitchell, Yi and Baum-yang05). To achieve this goal, we develop an ice cloud optical property parameterization scheme (referred to as the Voronoi scheme hereafter) for the Voronoi model. The Voronoi scheme and other schemes are employed in the Community Integrated Earth System Model (CIESM) (Lin et al., 2020) to simulate shortwave and longwave fluxes at the top of the atmosphere (TOA). The CIESM-based simulations from five schemes are compared with the Earth's Radiant Energy System (CERES) Energy Balanced And Filled (EBAF) products. This study addresses the following questions through the comparison. How different is the Voronoi scheme from other schemes? How, and to what extent does the Voronoi scheme affect the radiative effects of ice clouds? What are the possible reasons for the impacts of the Voronoi scheme?

This paper is organized as follows. Sections 2 and 3 introduce the data and methodology used in this study, section 4 demonstrates the influence of the Voronoi model on the cloud radiative properties through the radiative transfer model and climate model. Section 5 presents the summary and conclusion of this study.

## 2 Data

### 2.1 Single-scattering property database for the Voronoi model

In this study, the single-scattering property database of the Voronoi model developed by Ishimoto et al. (2012) and Letu et al. (2016) is used in the parameterization process. The single-scattering properties including the extinction efficiency, single-scattering albedo, and asymmetry factor of the Voronoi model from the ultraviolet to the infrared are utilized to calculate the shortwave and longwave optical properties of ice clouds. The single-scattering properties of the Voronoi model are calculated by combination methods of FDTD, GOIE and GOM. This combination method implements a well treatment of particle edge effects (Ishimoto et al., 2012). With this treatment, the gaps between the results calculated by the FDTD and those by the GOIE are relatively small, which can lead to consistence in the single-scattering properties of ice particles. Figure 1 (a) and (b) show the extinction efficiency, single-scattering albedo, and asymmetry factor for Voronoi models that vary with SZPs at wavelengths of 0.64 and 2.21 μm, respectively. Note that the FDTD and GOIE methods are used for small (SZP < 40) and moderate (SZP < 300) ice particles, respectively, and the GOM method is used for large (SZP > 300) particles. The extinction efficiency at wavelengths of 0.64 μm and 2.21 μm has a peak value when the SZP approximately equals to 10 and decreases to be a constant value of 2 with increasing SZPs larger than 100. The single-scattering albedo at both wavelengths is close to 1, which is related to the high values of the imaginary part in the refractive index. The asymmetry factor decreases with the increasing SZPs at the wavelength of 0.64 μm. At the wavelength of 2.21 μm, the asymmetry factor increases for the SZPs smaller than 10 and larger than 300.

### 2.2 Aircraft-based measurements of the particle size distributions

To generate the parameterization scheme of ice cloud optical properties for application in the climate model simulations, particle size distributions (PSDs) of the Voronoi model need to be assumed in ice clouds. In this study, we utilized 14,408 PSDs derived from in situ aircraft-based measurements obtained in 11 field campaigns (available http://stc-se.com/data/bbaum/Ice_Models/microphysical_data.html) (Heymsfield et al., 2013). These data confirm that the particle phase is unambiguously ice after filtering by cloud temperature (T≤-40°C). For the fitting of PSDs for the Voronoi model, we adopt the gamma distribution form as follows:

$$n(L) = N_0 L^\mu e^{-\lambda L} \tag{1}$$

where $L$ is the particle maximum dimension, $n(L)$ is the particle concentration per unit volume (e.g., 1/cm$^3$), $N_0$ is the intercept, $\lambda$ is the slope, and μ is the dispersion. The physical meaning of the PSD is that $n(L)$ times $dL$ is the number of particles per unit area.

As shown in Figure 2, the particle concentration decreases with increasing $L$ for all ranges. When the temperature is between -60 ℃ and -55 ℃ temperature, the particle concentration is the largest, and it decreases most sharply with increasing $L$. When the temperature is between -45 ℃ and -40 ℃, as well as between -65 ℃ and -70 ℃ temperature, the particle concentration is the smallest, and it decreases slowly with increasing $L$.

**2.3 Satellite data used in the validation**

To evaluate the cloud radiative effects for different schemes of the ICS model, we adopted the CERES EBAF Ed4.1 products (available https://ceres.larc.nasa.gov/data/) (Draine and Flatau, 1994; Doelling et al., 2016) from 2001 to 2010 as validation data. The "toa_cre_sw_mon" and "toa_cre_lw_mon" EBAF products are used to compared with the simulated shortwave and longwave cloud radiative effects from all five schemes. The "toa_cre_sw_mon" and "toa_cre_lw_mon" products are monthly mean shortwave and longwave cloud radiative effects at the TOA, and they are calculated as all-sky fluxes minus total region clear-sky fluxes for shortwave and longwave spectrum. The "toa_sw_all_mon" and "toa_lw_all_mon" EBAF products are monthly mean all-sky outgoing shortwave and longwave fluxes at the TOA, and they are used to compare with simulated upwelling shortwave flux at the TOA (FSUTOA) and upwelling longwave flux at the TOA (FLUTOA) from all schemes. The "sfc_sw_down_all_mon" and "sfc_lw_down_all_mon" are monthly mean all-sky downwelling shortwave and longwave fluxes at the surface, and they are used to compare with simulated downwelling shortwave flux at the surface (FSDS) and downwelling shortwave flux at the surface (FLDS) from all schemes. The spatial and temporal resolution of EBAF data is 1°×1° latitude by longitude and monthly means.

**3 Methodology**

The main flowchart of this study is described in Figure 3. Firstly, we developed the parameterization scheme of the Voronoi ICS model by using the aforementioned single-scattering properties of the Voronoi ICS database and large amounts

of PSDs. Then, the Voronoi scheme and the other four existing schemes (Fu, Mitchell, Yi and Baum-yang05) were evaluated

through simulations of shortwave upward/downward flux profiles in the Rapid Radiative Transfer Model for General

circulation model version (RRTMG). The RRTMG was utilized to understand how the different optical properties of the five

schemes influence the upward/downward fluxes under several idealized conditions. Furthermore, all schemes were applied

to the CIESM to simulate global shortwave and longwave cloud radiative forcing at the TOA from 2000 to 2010, among

which the first year was removed for reaching the equilibrium state and the last ten years were evaluated by EBAF products.

The CIESM was employed to evaluate the effectiveness of the Voronoi ICS model in the simulations of ice cloud radiative

properties compared with the other four schemes in the climate system.

**3.1 Parameterization of ice cloud optical properties**

To better understand the ice cloud modelling capabilities of the Voronoi model in climate model and explain how ice

clouds play a role in the climate system, it is necessary to introduce the main scattering parameters to evaluate the Voronoi

ICS model through the parameterization scheme. The main radiative transfer processes can be simply attributed to extinction,

scattering and absorption coefficients (Liou, 1986, 1992), which are calculated by Eq. (2).

$$\beta_{e,s,a} = \int_{L_{min}}^{L_{max}} \sigma_{e,s,a} n(L) dL \tag{2}$$

where $\beta_{e,s,a}$ is the extinction, scattering and absorption coefficients and $\sigma$ is the cross section (See Table 1 for a list of

acronyms). The single-scattering albedo and co-albedo can be defined as the ratio of the scattering and absorption

coefficients to the extinction coefficient in the form of Eq. (3).

$$\varpi = \frac{\beta_s}{\beta_e} \text{ , or } 1 - \varpi = \frac{\beta_a}{\beta_e} \tag{3}$$

where $\varpi$ and $1 - \varpi$ are single-scattering albedo and co-albedo, respectively. Based on the extinction coefficient, the

optical depth can be defined by Eq. (4).

$$\tau = \int_z^\infty \beta_e \, dz \tag{4}$$

where $\tau$ is the optical depth, $z$ is the outer boundary of the atmosphere. In the assumption of plane-parallel atmospheres, changes in the diffuse intensity penetrating from below the layer considering multiple scattering processes can be given by Eq. (5).

$$J(\tau; \mu; \phi) = \frac{\varpi}{4\pi} \int_0^{2\pi} \int_{-1}^{1} I(\tau; \mu'; \phi') P(\mu, \phi; \mu', \phi') d\mu' d\phi'$$

$$+ \frac{\varpi}{4\pi} F_\theta P(\mu, \phi; -\mu_0, \phi_0) e^{-\tau/\mu_0} + (1 - \varpi) B[T(\tau)]$$

(5)

where $P$ is the phase function corresponding to a volume of ice particles. $P$ (μ, φ; μ', φ') denotes the redirection of the incoming intensity defined by (μ', φ') to the outgoing intensity defined by (μ, φ). $I$ indicate the total (direct plus diffuse)

radiance, $B$ indicates Planck's function associated with thermal emissions, and $\Theta$ is the scattering angle. Therefore, the extinction coefficients, single-scattering albedo and phase function are fundamental driving parameters within the transfer of diffuse intensity.

Based on these principles, in this study, we completed the Voronoi scheme by using the single-scattering properties of the Voronoi model and 14,408 groups of PSDs data. The parameterization of the ice cloud optical properties for Voronoi

scheme are developed following Eq. (6)-(15). Firstly, the spectral ice cloud optical properties (mass-averaged extinction coefficients, single-scattering albedo and asymmetry factor) for the Voronoi scheme are calculated for all PSDs given by Eq. (6)-(9).

$$D_e = \frac{3}{2} \frac{\int_{L_{min}}^{L_{max}} V(L)n(L)dL}{\int_{L_{max}}^{L_{max}} A(L)n(L)dL}$$

(6)

$$K_{ext}(\lambda) = \frac{\int_{L_{min}}^{L_{max}} Q_{ext}(\lambda, L)A(L)n(L)dL}{\rho_{ice} \int_{L_{min}}^{L_{max}} V(L)n(L)dL}$$

(7)

$$\varpi(\lambda) = \frac{\int_{L_{min}}^{L_{max}} Q_{sca}(\lambda, L)A(L)n(L)dL}{\int_{L_{min}}^{L_{max}} Q_{ext}(\lambda, L)A(L)n(L)dL}$$

(8)

$$g(\lambda) = \frac{\int_{L_{min}}^{L_{max}} g(\lambda, L)\sigma_{sca}(\lambda, L)n(L)dL}{\int_{L_{min}}^{L_{max}} \sigma_{sca}(L)n(L)dL}$$

(9)

where $D_e$ is the effective particle diameter, $V$ and $A$ are volume and projected area of Voronoi models. $K_{ext}(\lambda)$ are spectral mass-averaged extinction coefficients (m²/g), $\varpi(\lambda)$ is spectral single-scattering albedo and $g(\lambda)$ is spectral asymmetry factor. $Q_{ext}, g$ and $Q_{sca}$ are extinction efficiency, asymmetry factor and scattering efficiency for Voronoi models.

Then, based on the spectral bulk optical properties including $K_{ext}(\lambda)$, $\varpi(\lambda)$, and $g(\lambda)$ of ice clouds, the band-averaged optical properties are calculated to apply the parameterization scheme in RRTMG and CIESM following Eq. (10)-(12).

$$\widetilde{K}_{ext} = \frac{\int_{\lambda_{min}}^{\lambda_{max}} \beta_{ext}(\lambda)E(\lambda)d\lambda}{\int_{\lambda_{min}}^{\lambda_{max}} E(\lambda)d\lambda} \tag{10}$$

$$\widetilde{\varpi} = \frac{\int_{\lambda_{min}}^{\lambda_{max}} \varpi(\lambda)E(\lambda)d\lambda}{\int_{\lambda_{min}}^{\lambda_{max}} E(\lambda)d\lambda} \tag{11}$$

$$\widetilde{g} = \frac{\int_{\lambda_{min}}^{\lambda_{max}} g(\lambda)E(\lambda)d\lambda}{\int_{\lambda_{min}}^{\lambda_{max}} E(\lambda)d\lambda} \tag{12}$$

where $\widetilde{K}_{ext}$, $\widetilde{\varpi}$ and $\widetilde{g}$ are band-averaged mass-averaged extinction coefficients, single-scattering albedo and asymmetry factor for the Voronoi scheme, respectively. $E$ is assigned by the solar constant provided by Chance and Kurucz (2010) for the shortwave spectrum, and is replaced with the Planck function $B(T)$ for longwave spectrum, T is an assuming cloud temperature of 233K according to Liou (1992). The coefficients of the polynomial expressions of the ice cloud band-averaged optical properties as functions of $D_e$ are determined in each band interval to develop Voronoi scheme for shortwave and longwave spectrum as shown in Eq. (13)-(15).

$$\widetilde{K}_{ext} = a_0 + a_1/D_e + a_2/D_e^2 \tag{13}$$

$$\widetilde{\varpi} = b_0 + b_1 D_e + b_2 D_e^2 + b_3 D_e^3 \tag{14}$$

$$\widetilde{g} = c_0 + c_1 D_e + c_2 D_e^2 + c_3 D_e^3 \tag{15}$$

where a, b, c are coefficients as functions of band intervals.

In terms of other four existing schemes, the band-averaged optical properties of Mitchell, Yi and Baum-yang05 schemes are developed as functions of $D_e$ following Eq. (13-15). Coefficients of Mitchell scheme can be obtained from the CIESM. Values of coefficients for Yi and Baum-yang05 schemes are listed in appendix A (Tables A1, A2, A3, and A4)

provided from Zhao et al. (2018). Coefficients of the Fu scheme (default scheme in RRTMG) are obtained from the existing ice cloud band-averaged optical properties from RRTMG. Formulation of Fu scheme is similar to the Mitchell and Baum-yang05 schemes except using the generalized effective diameter (Fu, 1996). The generalized effective diameter of the Fu scheme is unified into $D_e$ for comparability.

### 3.2 RRTMG and CIESM simulation experiments

The version of the RRTMG used in this study is the current version of the radiative transfer code applied in the CIESM (Mlawer et al., 1997; Iacono et al., 2008; Clough et al., 2005; available from http://rtweb.aer.com). RRTMG utilizes the correlated-k approach to calculate shortwave fluxes and heating rates efficiently and accurately for application to climate models. The version of RRTMG utilizes a two-stream method for radiative transfer calculation. RRTMG has 14 bands for shortwave spectrum and 16 longwave bands (see Table 1). Since the wavelength range is from 0.2 to 15 µm for the single-scattering property database of the Voronoi ICS model, ice cloud bulk optical properties of the default scheme (the Mitchell scheme) remain unchanged when bands are larger than 15 µm. To quantify the radiative flux differences caused by five schemes under the same conditions, we design an assuming ice cloud cases in standard tropics atmospheric profile in the RRTMG. The RRTMG sets are as follows, ice effective radius is set to 45 µm, the ice water path is set to 60 g m$^{-2}$, the ice cloud top pressure/height is between 125.1 hPa and 245.5 hPa, and the cloud fraction is 50%, the solar zenith angles is set to 60 °. The vertical resolution is 60 levels for the standard tropics. The RRTMG is run by five ice cloud schemes under the same conditions, thus relative difference of fluxes can be explained by difference among five schemes. Five schemes are implemented in the CIESM to calculate the FSDS, FLDS, FSUTOA, FLUTOA and shortwave cloud forcing (SWCF) and longwave cloud forcing (LWCF). SWCF are defined as Eq. (16), LWCF is defined the same as SWCF but for longwave spectrum.

$$SWCF = F_{cloudy} - F_{clear} \tag{16}$$

where $F_{cloudy}$ and $F_{clear}$ are the difference between downward and upward fluxes for cloudy and clear conditions, respectively. The CIESM is run in two ways: 1) the CIESM is run with the default Mitchell scheme for ice clouds and the default water cloud scheme to obtain SWCF and LWCF for the Mitchell scheme; 2) the CIESM is run by using the other four schemes (the Voronoi, Yi, Baum-yang05 and Fu scheme) in place of the Mitchell scheme, along with the default liquid

water cloud scheme. Liquid water clouds adopt a spherical particle model, whose single-scattering properties are derived from the Lorenz-Mie theory (van de Hulst, 1957). Because the CIESM is unable to separate ice clouds from liquid clouds efficiently, the SWCF and LWCF for five schemes are under the same liquid water cloud parameterization their differences are attributed to different ice habits and their scattering and absorption properties within five schemes. The CIESM run is integrated for 11 years in one-month increments, the initial first year is used for state initialization and stabilization, the last ten-year runs were utilized for analysis. Horizontal and vertical resolution of CIESM run experiment is set to 1.9° × 2.5° and 31 levels, respectively. The run is driven by prescribed climatological sea surface temperature and sea ice fraction with an annual cycle in the year 2000.

## 4 Results and discussions

### 4.1 Band-averaged optical properties of the ice cloud

Based on the integration over both PSDs and band intervals, band-averaged bulk optical properties of the Voronoi scheme are compared with Mitchell, Fu, Baum-yang05 and Yi schemes in Figure 4. The differences in ice cloud optical properties between the Voronoi and the other four existing schemes are shown in Figure 5. Since Fu scheme uses generalized effective diameter, the remaining four schemes use $D_e$. Horizontal axes in both Figure 4 and 5 are unified into $D_e$ for comparability. Parameterized mass extinction coefficients, single-scattering albedo and asymmetry factor are plotted as functions of the $D_e$ from 10 to 150 μm and 14 selected bands.

In Figure 4, mass extinction coefficients obtained from five schemes show uniformly negative correlation with $D_e$. Mass extinction coefficients exceed up to 0.2 m²/g for $D_e$ smaller than 20 μm and approach 0 for $D_e$ larger than 100 μm. This could be partly related with the majority of small particles in PSDs for five schemes. Note that there is a minimum of mass extinction coefficients between 3.08 and 3.85 μm. It could because that the real part of the refractive index reaches the minimum at 3 μm (Warren and Brandt, 2008; Yang et al., 2013). This could weaken the scattering and extinction efficiency of ice particles. The single-scattering albedos obtained from five schemes approach 1.0 in visible bands (0.2-0.78 μm). In near-infrared bands (0.78-3.85 μm), the single-scattering albedos are inversely proportional to $D_e$. This result is related with the large real part in visible bands and small imaginary part of the complex refractive index of ice particles. The

asymmetry factor obtained from five schemes increases with increasing wavelength for all $D_e$. From visible to near-infrared band, the asymmetry factor increases with the increasing $D_e$.

In Figure 5, differences in mass extinction coefficients between the Voronoi scheme and the other four schemes show the Voronoi scheme has slightly larger values than the Fu, Yi and Baum-yang05 schemes, but is smaller than the Mitchell scheme.     For the single-scattering albedo, the Voronoi scheme has slightly larger single-scattering albedo than Fu and Mitchell schemes, and lower single-scattering albedo than Baum-yang05 and Yi schemes in infrared bands. This result may be because large ice particles are closer to geometric optics and have a greater proportion of absorption than small ice particles. The low asymmetry factor of the Voronoi scheme is because that the multifaceted shapes of the Voronoi ice model can result in significant side and backward scattering and reducing the forward scattered energy. Since the impacts of different size distribution assumptions on the bulk optical properties of ice cloud parameterization are negligible (Heymsfield et al., 2013, 2017), differences of band-averaged bulk optical properties between five schemes are originally rooted in different habits of ice particles and their single-scattering properties.

## 4.2 RRTMG simulation results

After the parameterization, band-averaged optical properties of ice cloud from five schemes (Fu, Mitchell, Yi, Baum-yang05 and Voronoi) are subsequently parameterized as functions of $D_e$ and 14 bands as shown in Figure 4. To illustrate and quantify the influence of optical properties of ice cloud on its radiative effects, an ideal experiment is designed to test the response of radiative flux to five ice cloud schemes under the same idealized conditions. Band-averaged optical properties for five schemes are subsequently implemented in RRTMG to simulate radiative fluxes under prescribed ice clouds in standard tropics profiles (Anderson et al., 1986) which have a high proportion of ice cloud coverage (Massie et al., 2002; Stubenrauch et al., 2013). According to observations of Hong and Liu (2015), top and bottom pressure of ice cloud layer is set to 125.1 and 245.5 hPa, respectively, the $D_e$ is set to 45 μm and ice water paths equal to 60 g m$^{-2}$.

Shortwave radiative fluxes profiles of cloudy-sky for five schemes and clear-sky conditions are shown in Figure 6. Specific comparison of five schemes inside the black dotted region are enlarged and shown in the bottom row. In Figure 6, upward fluxes of five schemes gradually increase from cloud bottom to cloud top, reaching to the maximum at the cloud top. The Voronoi and Mitchell scheme have higher upward fluxes and lower downward diffuse fluxes than the other three

schemes. Figure 6 shows 6-30 W/m$^2$ differences in TOA upward fluxes, 10-40 W/m$^2$ differences in surface downward diffuse fluxes, 10-30 W/m$^2$ differences in surface net fluxes, and 8-42 W/m$^2$ differences in TOA net fluxes owing to five different ice cloud schemes. The radiative properties for the Voronoi scheme in shortwave fluxes can be explained by its

lower asymmetry factor than the other four schemes, leading to smaller proportion of forward scattering and larger backward scattering. Thus, less shortwave flux reaching the ground and more upward flux for the Voronoi scheme compared with the other four schemes.

### 4.3 Climate model simulation results

As shown in RRTMG simulations in section 4.2, the influence of five ice cloud schemes (Voronoi, Yi, Mitchell, Baum-

yang05 and Fu schemes) on the radiative effects is evaluated under standard tropical atmospheric profiles, and with assumptions of idealized ice cloud microphysical properties as input data. Simulation results based upon radiative transfer model are capable of showing difference of ice cloud radiative effects for five schemes under some specific conditions, but are unable to demonstrate comprehensive performance of five schemes corresponding to real and complex atmospheric situation. To study the ice cloud modelling capabilities of five schemes in the climate model, the Voronoi, Yi, Baum-yang05

and Fu schemes are applied in the CIESM. Upwelling and downwelling fluxes and TOA SWCF and LWCF from CIESM simulations for five schemes compared with CERRES EBAF products are plotted in Figure 7 and 8, and their global-averaged values are listed in Table 3. The results show that the Voronoi scheme produced a lower difference of approximately -0.45 W/m$^2$ (1.1%) for the TOA shortwave cloud radiative forcing and -0.30 W/m$^2$ (1.4%) for the TOA longwave cloud radiative forcing compared with four existing schemes. For FSDS, the Voronoi scheme has the smallest

downwelling fluxes at surface and is the closest to the EBAF products due to that the Voronoi scheme scatter the least energy in the forward direction. For FSUTOA, the Voronoi scheme possesses the largest upwelling fluxes compared to the other four schemes due to its strong backward scattering. For the longwave spectrum, the effects of the Voronoi scheme on the FLDS and FLUTOA is negligible.

To discuss the influence of five schemes on the global distributions of SWCF and LWCF, the zonal average analysis is

shown in Figure 9. Results shows that the Voronoi scheme exhibits weaker cooling effects in tropical regions than the other four existing schemes. According to the definitions of the SWCF and LWCF, the highest TOA upward fluxes of the Voronoi

scheme can produce the lowest TOA net fluxes, which means the Voronoi scheme can produce the smallest negative TOA SWCF among five schemes. Figure 10 displayed the distribution of differences between five schemes and CERES EBAF product. The differences box of Voronoi scheme are most concentrated on the zero line, and its statistical deviation is the smallest, which means the spatial distribution of cloud radiative effects of Voronoi scheme is closer to EBAF products compared with other four existing schemes.

## 5 Conclusions

The optical property parameterization (Voronoi scheme) of the Voronoi ice crystal scattering (ICS) model is investigated for simulations of the ice cloud radiative properties in the Community Integrated Earth System Model (CIESM). The Voronoi scheme is completed based on the single-scattering properties of the Voronoi ICS database and particle size distributions from in-situ observations. The band-averaged optical properties of ice clouds including the mass extinction coefficients, single-scattering albedo and asymmetry factor of the Voronoi scheme are applied in the CIESM climate model and compared with those of the four existing schemes (Baum-yang05, Fu, Yi and Mitchell). The results show that the Voronoi scheme has a distinct feature of having the lowest asymmetry factor in the shortwave bands. This feature could be related to complex multifaceted shape of the Voronoi ICS model, and suggests that the Voronoi scheme can produce relatively stronger backward scattering compared with other schemes.

Radiative properties of ice clouds are firstly assessed in Rapid Radiative Transfer Model for General circulation model version (RRTMG) in the CIESM. The profiles of upward/downward fluxes from different ice cloud schemes are simulated for the prescribed atmospheric condition. The RRTMG results show that the Voronoi scheme has the highest upward flux at the top of the atmosphere (TOA) and lowest downward flux at the surface when the solar zenith angle equals to 60 °. Therefore, the net flux of the Voronoi scheme is largest at the TOA and smallest at the surface compared with the other schemes, which are mostly due to its lowest asymmetry factor.

Five schemes (Baum-yang05, Fu, Yi, Mitchell and Voronoi) are then applied to the CIESM to simulate shortwave and longwave global total cloud radiative forcing at the TOA during 2001-2010. The accuracy of simulated 10-yr global total cloud radiative forcing from different schemes are evaluated by the Clouds and the Earth's Radiant Energy System (CERES)

Energy Balanced And Filled (EBAF) product. The results show that the Voronoi scheme produced a lower difference for the TOA shortwave and longwave cloud radiative forcing compared with four existing schemes. Especially for the region (from 30°S to 30°N) where the ice clouds occur frequently, the Voronoi scheme provides the closest match with CERES EBAF product.

In conclusion, simulations of global averaged shortwave and longwave cloud radiative forcing at the TOA from five schemes are investigated through the EBAF product. We find that the Voronoi scheme present a better agreement with EBAF products than the other schemes, especially in the tropical region, which confirmed that the Voronoi ICS model has the possibility of ice cloud modelling capabilities in the climate model of the CIESM.

**Data availability**

The RRTMG code are available at http://rtweb.aer.com/rrtmg_sw_code.html, the CERES level3 EBAF products are available at https://ceres.larc.nasa.gov/data/.

**Author contribution**

Ming Li developed the ice cloud optical property parameterizations (Voronoi scheme) based on the single-scattering properties of Voronoi models, compared the band-averaged optical properties of the Voronoi scheme with the other four

schemes (Mitchell, Yi, Baun-yang05 and Fu). Ming Li also compared the upward/downward flux profiles from five schemes through RRTMG standalone simulations and radiative properties of five schemes in CAM5 model simulations, as well as downloaded the CERES products and wrote the initial draft of this manuscript. Husi Letu designed the aims and structures of this study and assisted in developing the parameterization of ice cloud optical properties based on the Voronoi models. Husi Letu also provided the single-scattering property database of Voronoi models and helped in analyzing the single-scattering

properties of Voronoi models, as well as guided the writings and revisions of the manuscript. Yiran Peng and Yanluan Lin assisted in developing the ice cloud optical property parameterization and provided the climate models, as well as guided the settings of climate model runs and reviewing the manuscript. Hiroshi Ishimoto developed the single-scattering property database of Voronoi models, provided the database of Voronoi models and helped in the parameterization of ice cloud

optical properties based on the single-scattering properties of Voronoi models. Takashi Y. Nakajima provided the single-scattering property database of Voronoi models, especially assisted in guiding the flowchart of this study and reviewed the manuscript. Anthony Baran guided the development of the ice cloud optical property parameterization and reviewed the paper. Zengyuan Guo assisted with the runs and design of the climate model simulations and helped with the review of the manuscript. Yonghui Lei assisted in analyzing the results and guided the flowchart of the study, as well as reviewed the manuscript. Jiancheng Shi assisted in designing the aims and structures of this study, guided the writings of the paper and helped reviewing the manuscript.

## Competing interests

The authors declare that they have no conflict of interests.

## Acknowledgement

This work is supported by the Second Tibetan Plateau Scientific Expedition and Research Program (STEP) (Grant No. 2019QZKK0206), National Natural Science Foundation of China (Grant No. 42025504, 41771395) and International Partnership Programme of Bureau of International Cooperation Chinese Academy of Sciences (Contract No. 181811KYSB20190014).

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

## Table Caption:

Table 1. Nomenclature

Table 2. Shortwave and longwave bands set in the RRTMG

Table 3. TOA upwelling and downwelling surface fluxes (W m$^{-2}$) and TOA SWCF and LWCF (W m$^{-2}$) for five schemes compared with CERRES EBAF products. Numbers in parentheses are differences between five scheme simulations and CERES EBAF products.

## Table 1. Nomenclature

| | |
|---|---|
| $L$ | Particle maximum diameter ($\mu m$) |
| $\lambda$ | Wavelength ($\mu m$) |
| $SZP$ | Size parameter (unitless) |
| $n(L)$ | Particle concentration (cm$^{-3}$) |
| $N_0$ | Intercept coefficient of $n(L)$ (unitless) |
| $\lambda^*$ | Slope coefficient of $n(L)$ (unitless) |
| $\mu$ | Dispersion coefficient of $n(L)$ (unitless) |
| PSD | Particle size distribution defined by $N_0$, $\lambda^*$, $\mu$ and $L$ |
| TOA | Top of atmosphere |
| $\beta_{e,s,a}$ | Extinction, scattering and absorption coefficients |
| $\sigma_{e,s,a}$ | Extinction, scattering, absorption cross section |
| $\theta$ | Inclination to the upward normal direction scattering angle |
| $\mu$, $\mu$' | Cosines of $\theta$, incoming and outgoing intensity direction, respectively |
| $\varphi$, $\varphi$' | Incoming and outgoing intensity azimuthal angle in reference to the $x$ axis, respectively |
| $P$ | Phase function regulated by $\mu, \phi, \mu', \phi'$ |
| $z$ | Upper limit of the outer boundary |
| $\tau$ | Optical thickness |
| $I$ | Total (direct plus diffuse) radiance |
| $B[T]$ | Planck's function |
| $J(\tau; \mu; \varphi)$ | Source function |
| $D_e$ | Effective particle diameter |
| $Q_{ext,sca,}(\lambda, L)$ | Extinction efficiency and scattering efficiency |
| $V(L)$ | Ice particle volume ($\mu m^3$) |
| $A(L)$ | Average geometrical cross section ($\mu m^2$) |
| $K_{ext}(\lambda), \widetilde{K}_{ext}$ | Spectral and band-averaged of mass extinction coefficients |

| | |
|---|---|
| $\varpi(\lambda), \widetilde{\varpi}$ | Spectral and band-averaged single-scattering albedo |
| $g(\lambda), \tilde{g}$ | Spectral and band-averaged asymmetry factor |
| $N$ | Cloud fraction |
| $F_{cloudy}$ | Net fluxes of cloudy conditions |
| $F_{clear}$ | Net fluxes of clear conditions |
| FSDS | Downwelling shortwave flux at the surface |
| FLDS | Downwelling longwave flux at the surface |
| FSUTOA | Upwelling shortwave flux at the top of atmosphere |
| FLUTOA | Upwelling longwave flux at the top of atmosphere |
| SWCF | Shortwave cloud forcing |
| LWCF | Longwave cloud forcing |


Table 2. Shortwave and longwave bands in the RRTMG

| Shortwave | | Longwave | |
|---|---|---|---|
| Band | μm | Band | cm$^{-1}$ |
| 16 | 3.08–3.85 | 1 | 10–350 |
| 17 | 2.5–3.08 | 2 | 350–500 |
| 18 | 2.15–2.5 | 3 | 500–630 |
| 19 | 1.94–2.15 | 4 | 630–700 |
| 20 | 1.63–1.94 | 5 | 700–820 |
| 21 | 1.3–1.63 | 6 | 820–980 |
| 22 | 1.24–1.3 | 7 | 980–1080 |
| 23 | 0.78–1.24 | 8 | 1080–1180 |
| 24 | 0.63–0.78 | 9 | 1180–1390 |
| 25 | 0.44–0.63 | 10 | 1390–1480 |
| 26 | 0.34–0.44 | 11 | 1480–1800 |
| 27 | 0.26–0.34 | 12 | 1800–2080 |
| 28 | 0.2–0.26 | 13 | 2080–2250 |
| 29 | 3.85–12.2 | 14 | 2250–2380 |
| | | 15 | 2380–2600 |
| | | 16 | 2600–3250 |


Table 3. TOA upwelling and downwelling surface fluxes (W m$^{-2}$) and TOA SWCF and LWCF (W m$^{-2}$) for five schemes compared with CERRES EBAF products. Numbers in parentheses are differences between five scheme simulations and CERES EBAF products.

| | CERES EBAF | Mitchell scheme | Voronoi scheme | Yi scheme | Baum-yang05 scheme | Fu scheme |
|---|---|---|---|---|---|---|
| FSDS | 161.54 | 163.3 (1.76) | 162.13 (0.59) | 164.11 (2.57) | 164.48 (2.94) | 164.26 (2.72) |
| FLDS | 309.98 | 298.9 (-11.08) | 298.37 (-11.61) | 298.11 (-11.87) | 298.62 (-11.36) | 299.31 (-10.67) |
| FSUTOA | 102.20 | 102.79 (0.59) | 104.79 (2.59) | 102.65 (0.45) | 102.58 (0.38) | 100.36 (-1.84) |
| FLUTOA | 222.52 | 217.98 (-4.54) | 218.3 (-4.22) | 217.67 (-4.85) | 218.38 (-4.14) | 218.7 (-3.82) |
| SWCF | -42.52 | -43.73 (-1.21) | -42.97 (-0.45) | -44.67 (-2.15) | -45.55 (-3.03) | -46.66 (-4.14) |
| LWCF | 20.88 | 20.21 (-0.67) | 20.58 (-0.30) | 20.09 (-0.79) | 20.11 (-0.77) | 19.86 (-1.02) |

# Figure Captions:

Figure 1. Single-scattering properties (extinction efficiency, single-scattering albedo and asymmetry factor) of Voronoi model from the composite method based on the FDTD, GOIE, and GOM methods at the wavelengths of (a) 0.64 µm and (b) 2.21µm.

Figure 2. Variations of ice particle size distributions for different temperature.

Figure 3. Flowchart of the investigation of ice cloud modelling capabilities for the irregularly shaped Voronoi models in climate simulations

Figure 4. The comparison of (top row) mass extinction coefficients, (middle row) single-scattering albedo and (bottom row) asymmetry factor as functions of effective diameter and 14 shortwave bands for (a) Voronoi, (b) Mitchell, (c) Fu, (d) Baum-yang05, and (e) Yi schemes.

Figure 5. The (a) Mitchell, (b) Fu, (c) Baum-yang05 and (d) Yi schemes minus the Voronoi scheme 590    differences (%) in (top row) mass extinction coefficients, (middle row) single-scattering albedo and (bottom row) asymmetry factor as functions of effective diameter and 14 shortwave bands.

Figure 6. Shortwave (a) upward fluxes, (b) diffuse downward fluxes, (c) downward total fluxes and (d) net fluxes for the Voronoi, Mitchell, Fu, Baum-yang05, Yi schemes and clear conditions (blue line) for 595    standard tropical atmospheric profile. Graphics in black dotted box are magnified and displayed in (e)-(h).

Figure 7. The 10-yr global average CIESM-based TOA SWCF simulations for Voronoi, Mitchell, Fu, Baum-yang05 and Yi schemes and corresponding CERES EBAF products during 2001-2010.


Figure 8. Same as the Figure 7, but for TOA LWCF.

Figure 9. Comparison of 10-yr zonally averaged annual mean (a) SWCF and (b) LWCF between the Voronoi, Mitchell, Fu, Baum-yang05, Yi schemes and CERES EBAF. EBAF minus five schemes

differences of (c) SWCF and (d) LWCF.

Figure 10. Box analysis of zonal distributions of 10-yr annual mean SWCF (left) and LWCF (right) difference between the Mitchell, Voronoi, Fu, Baum-yang05, Yi scheme and EBAF products, respectively.


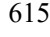

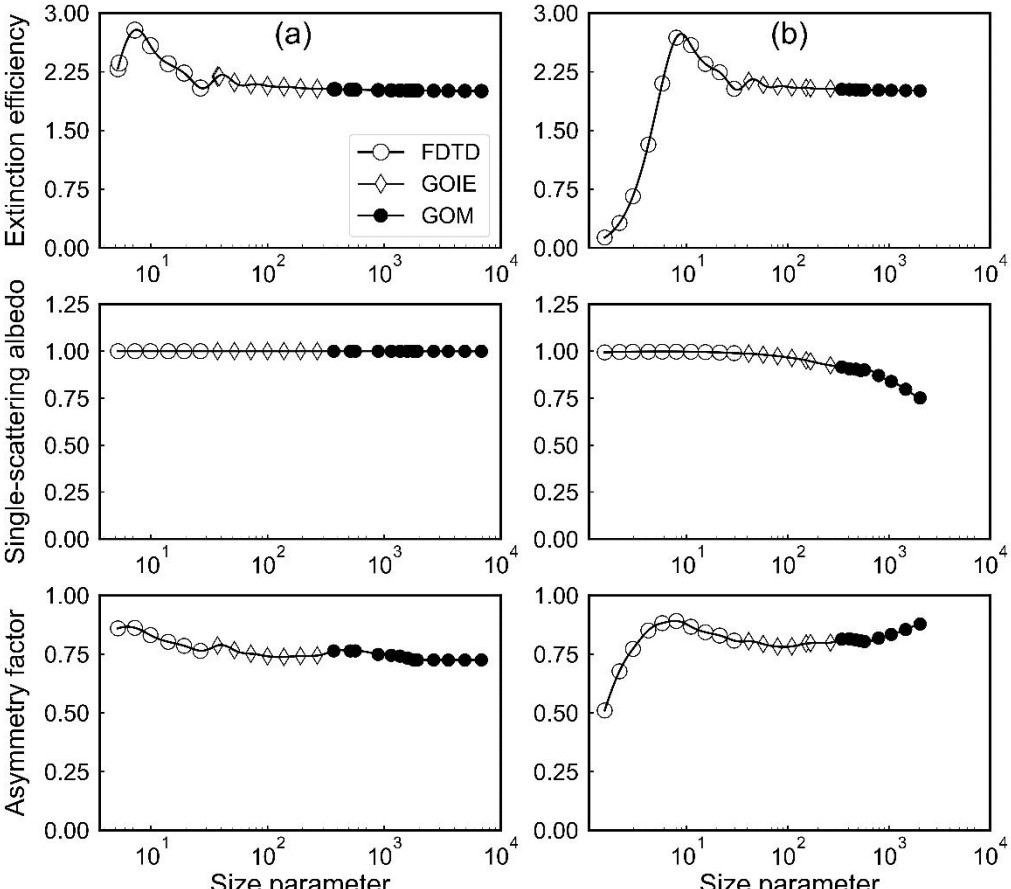

Figure 1. Single-scattering properties (extinction efficiency, single-scattering albedo and asymmetry factor) of Voronoi model from the composite method based on the FDTD, GOIE, and GOM methods at the wavelengths of (a) 0.64 μm and (b) 2.21μm.


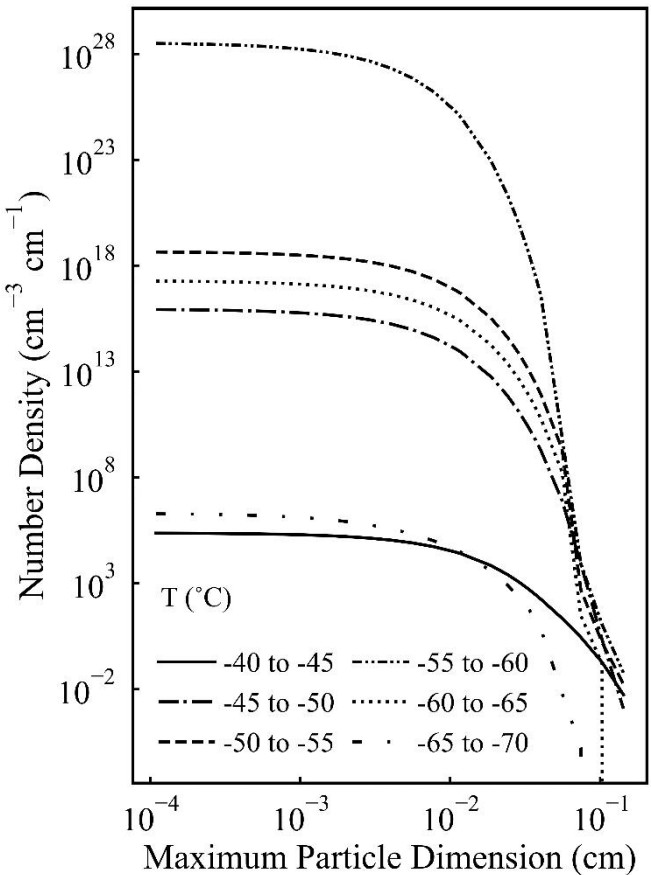

Figure 2. Variations of ice particle size distributions for different temperature.


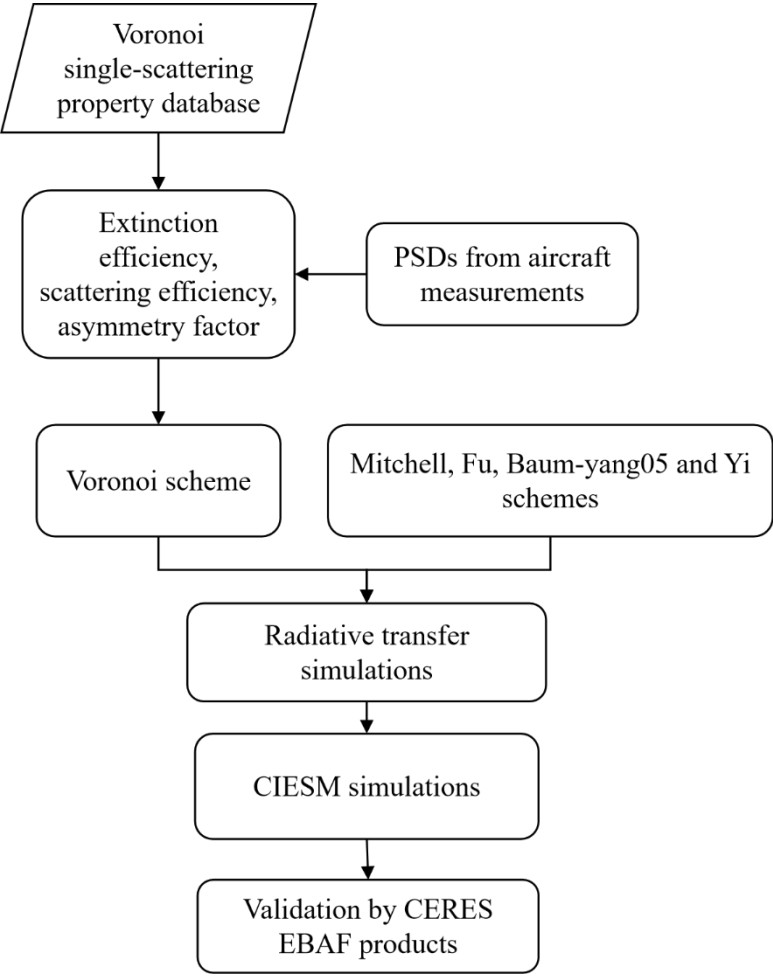

Figure 3. Flowchart of the investigation of ice cloud modelling capabilities for the irregularly shaped Voronoi models in climate simulations.

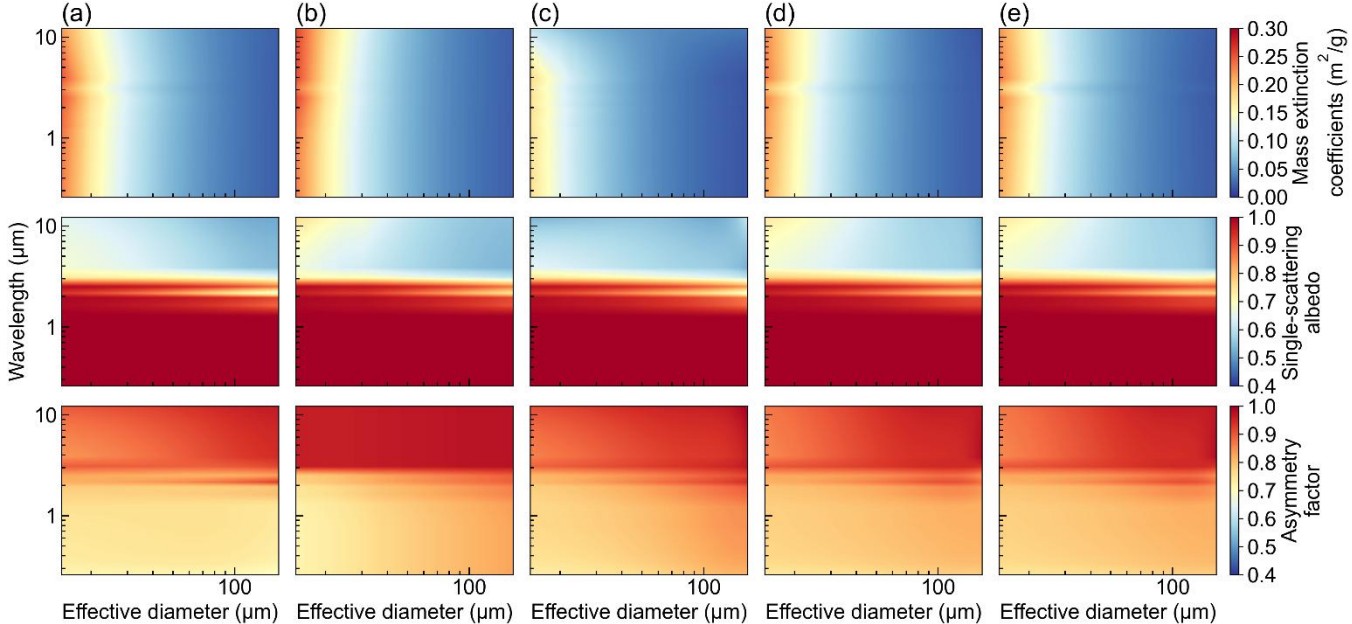

Figure 4. The comparison of (top row) mass extinction coefficients, (middle row) single-scattering albedo and (bottom row) asymmetry factor as functions of effective diameter and 14 shortwave bands for (a) Voronoi, (b) Mitchell, (c) Fu, (d) Baum-yang05, and (e) Yi schemes.

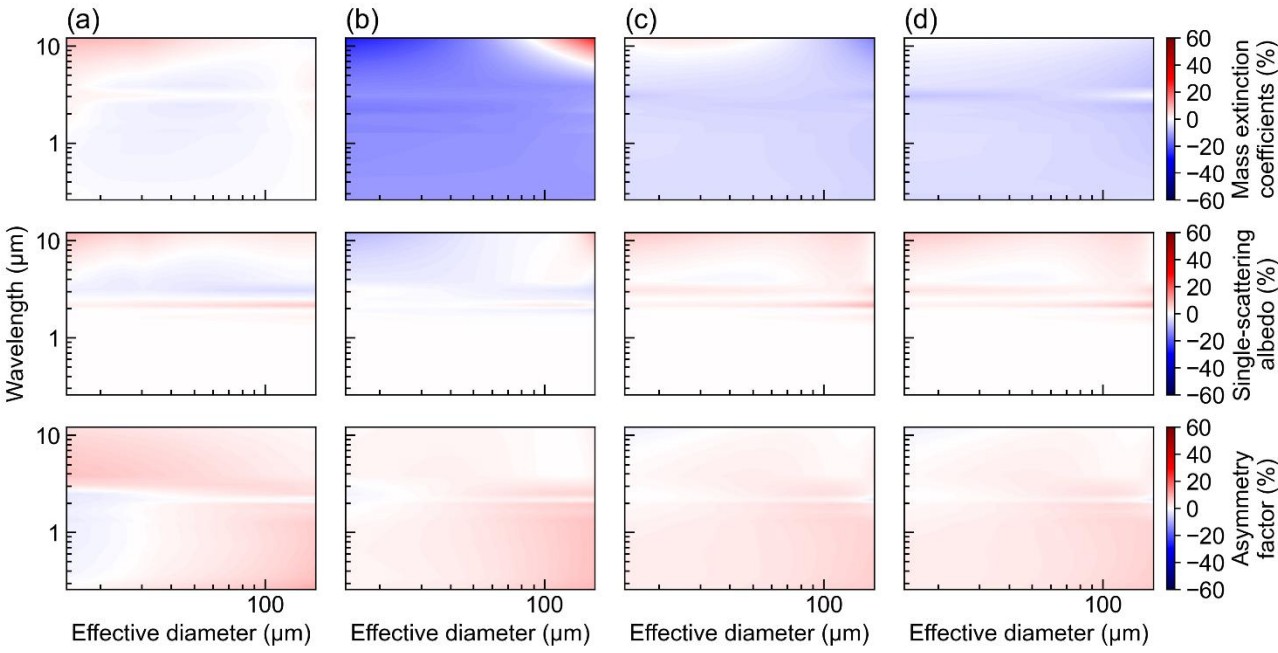


Figure 5. The (a) Mitchell, (b) Fu, (c) Baum-yang05 and (d) Yi schemes minus the Voronoi scheme

differences (%) in (top row) mass extinction coefficients, (middle row) single-scattering albedo and

(bottom row) asymmetry factor as functions of effective diameter and 14 shortwave bands.

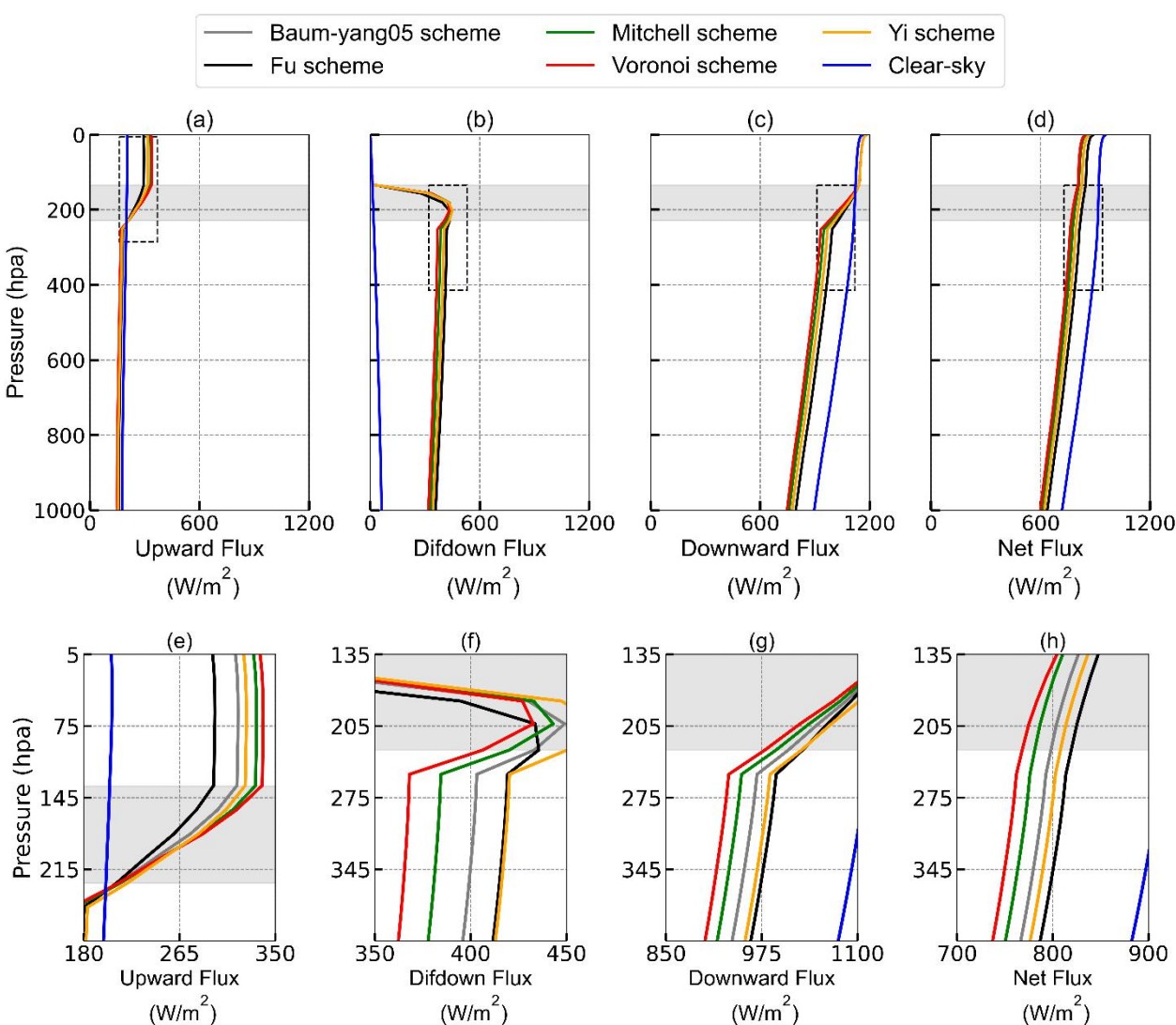

Figure 6. Shortwave (a) upward fluxes, (b) diffuse downward fluxes, (c) downward total fluxes and (d) net fluxes for the Voronoi, Mitchell, Fu, Baum-yang05, Yi schemes and clear conditions (blue line) for standard tropical atmospheric profile. Graphics in black dotted box are magnified and displayed in (e)-(h).

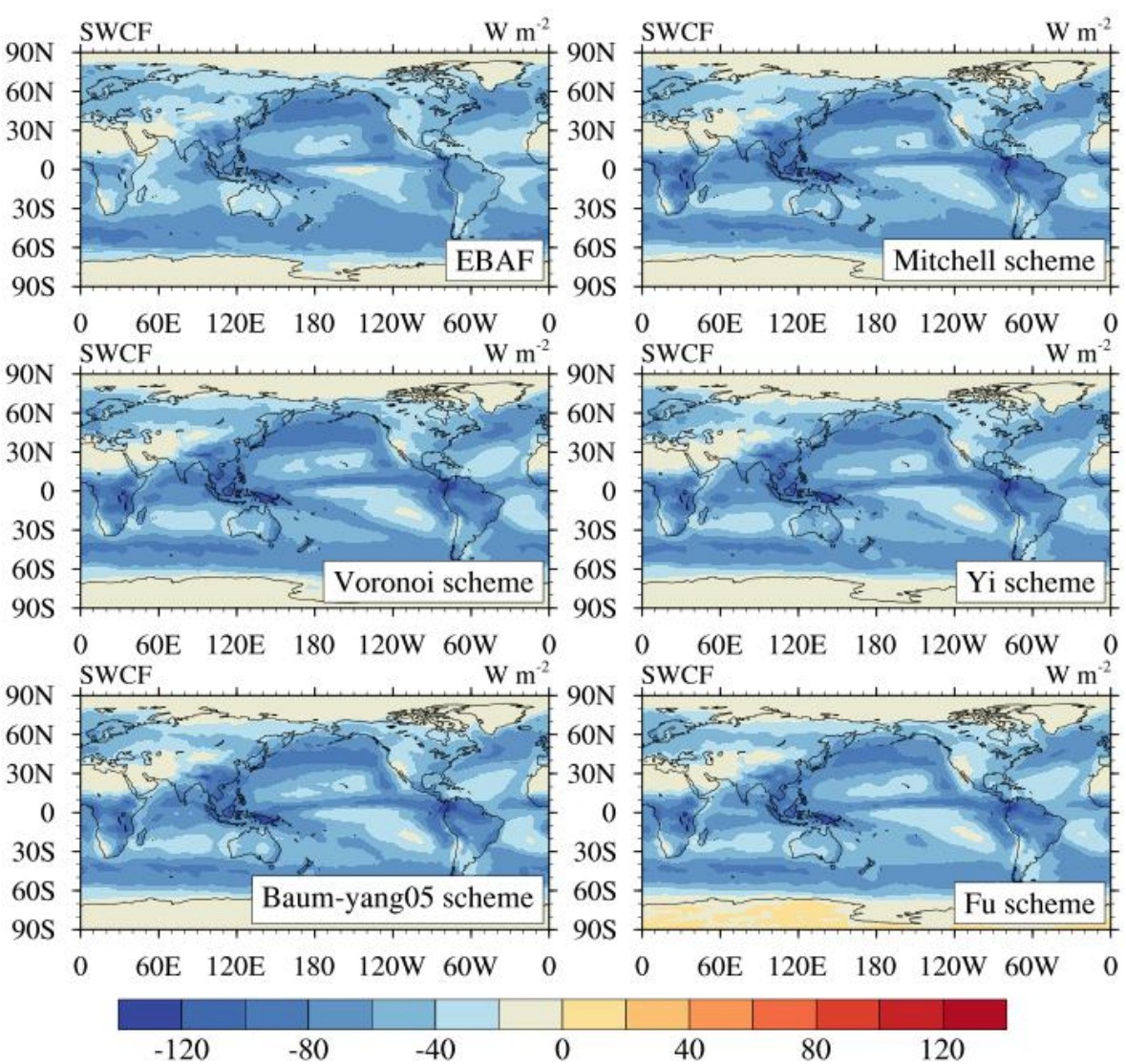

Figure 7. The 10-yr global average CIESM-based TOA SWCF simulations for Voronoi, Mitchell, Fu,

 Baum-yang05 and Yi schemes and corresponding CERES EBAF products during 2001-2010.

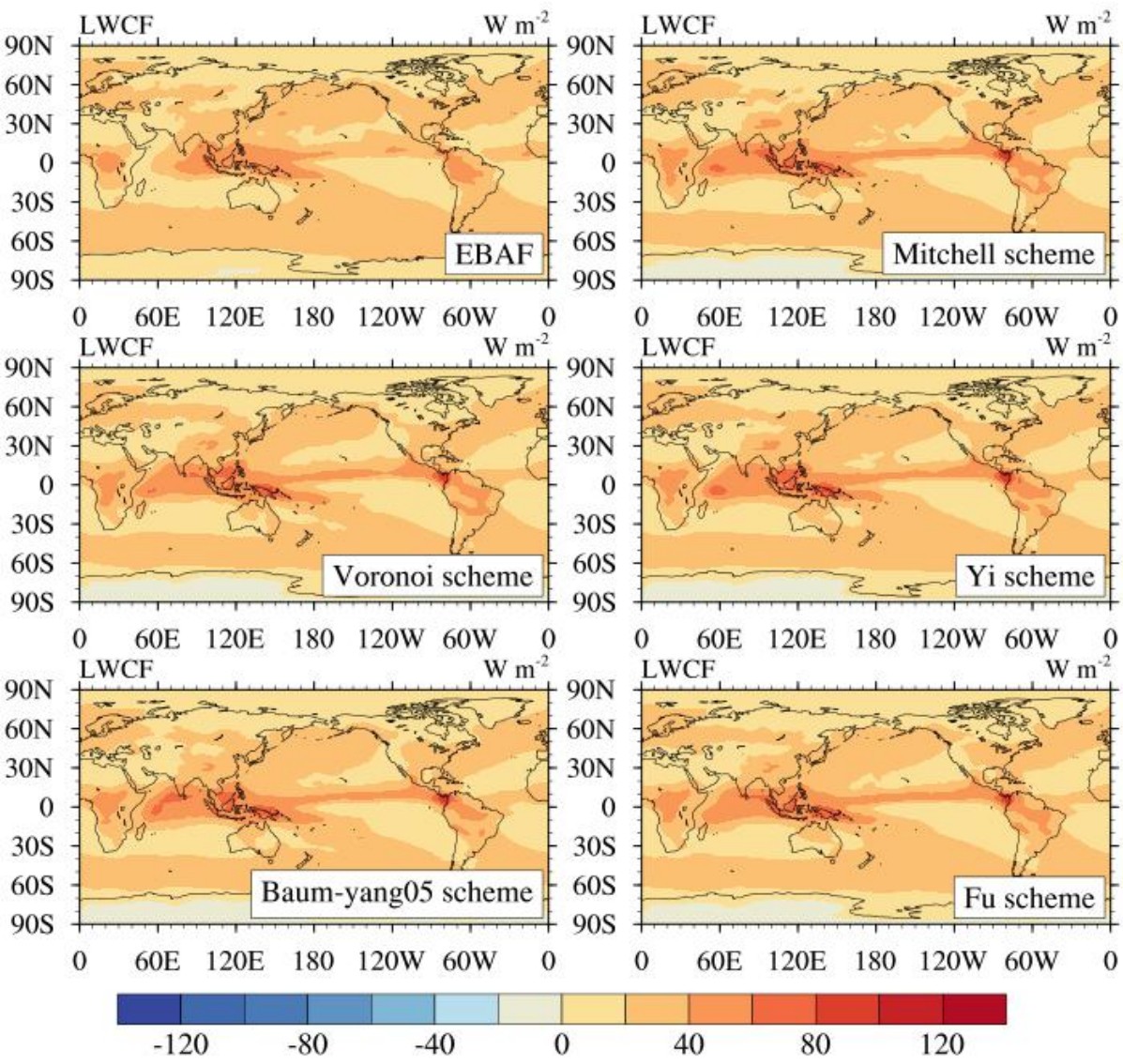

Figure 8. Same as the Figure 7, but for TOA LWCF.

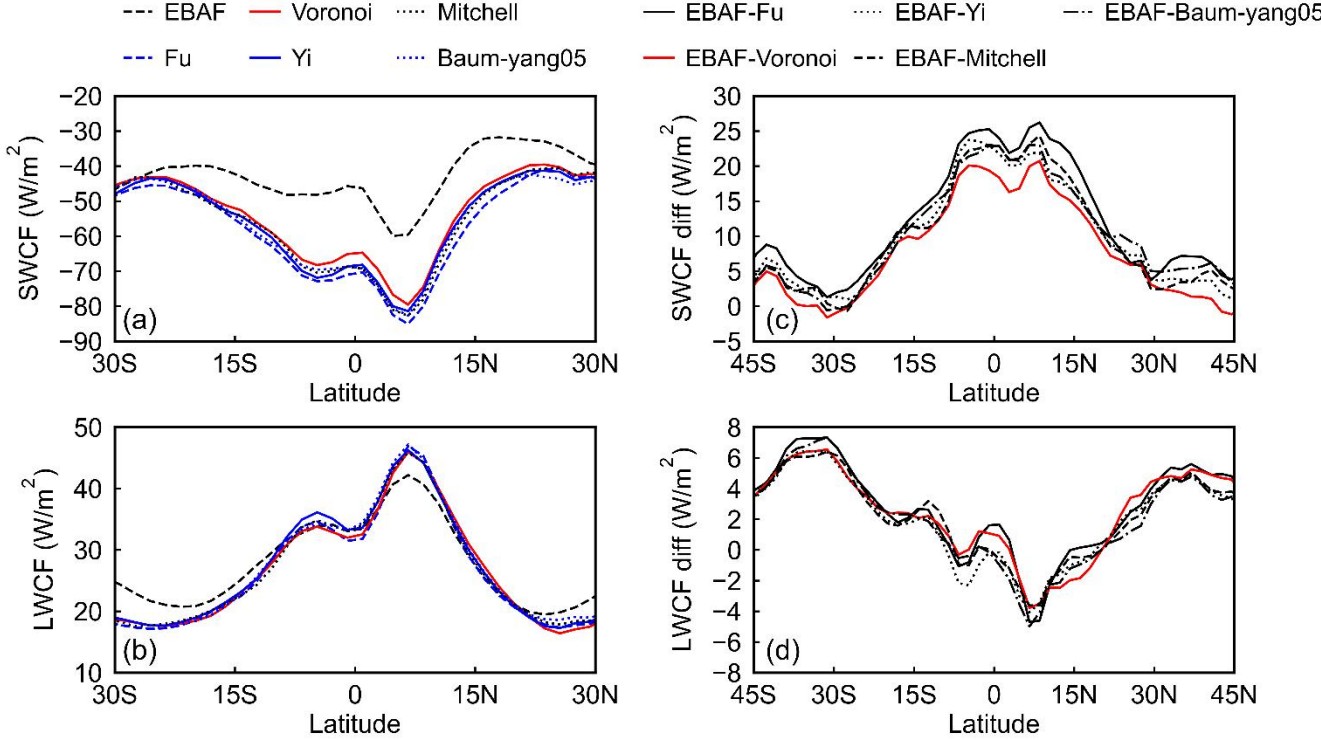

Figure 9. Comparison of 10-yr zonally averaged annual mean (a) SWCF and (b) LWCF between the Voronoi, Mitchell, Fu, Baum-yang05, Yi schemes and CERES EBAF. EBAF minus five schemes differences of (c) SWCF and (d) LWCF.

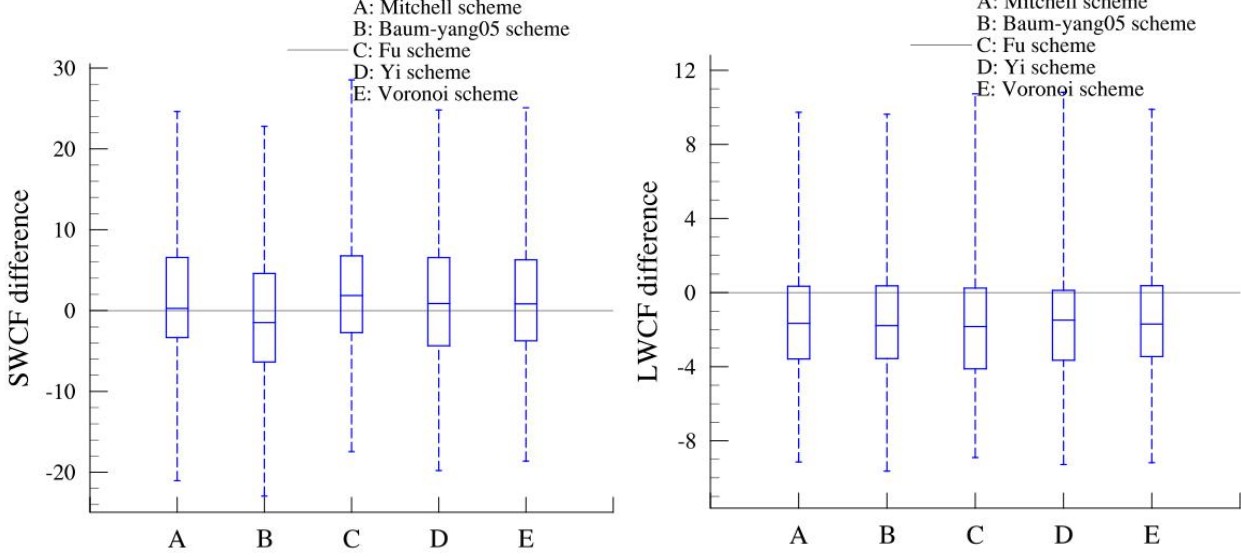

Figure 10. Box analysis of zonal distributions of 10-yr annual mean SWCF (left) and LWCF (right) difference between the Mitchell, Voronoi, Fu, Baum-yang05, Yi scheme and EBAF products, respectively.