# Peer review of "Investigation of ice cloud modelling capabilities for the irregularly shaped Voronoi ice scattering models in climate simulations"

_Atmospheric Chemistry and Physics, 2021_

## Referee Comment (RC1)

Manuscript number: acp-2021-208

Full title: Assessment of ice cloud modeling capabilities for the irregularly shaped Voronoi models in climate simulations with CAM5

Author(s): Li et al.

The paper assesses the performance of the Voronoi ice crystal model on the broadband radiative transfer simulations as well as climate simulations with CAM5 through the comparisons to other four ice cloud parameterizations including Mitchell, Fu, Baum–Yang, and Yi schemes. The Voronoi scheme exhibits relatively lower asymmetry factor and higher single-scattering albedo in the visible to near-infrared wavelength domain, than the other four schemes, resulting in more reflective ice clouds in shortwave radiative transfer simulations. The comparisons of the net cloud radiative effects between CERES observations and 10-yr CAM5 simulations among these ice cloud parameterization schemes suggest that the Voronoi scheme outperforms the other four parameterization schemes. The authors conclude that the Voronoi scheme can minimize the differences of the global TOA SWCF between the satellite-based measurements and the CAM5 simulation counterparts compared to other four schemes. This paper sufficiently describes the background and introduction, and methods. However, the result section contains inadequate discussions in the interpretation of the results. In particular, the authors should add more detailed descriptions on the five parameterization schemes. Also, there are numerous grammatical/language errors, and several sentences need to be rephrased. The topic presented in this study is suitable for Atmospheric Chemistry and Physics, and therefore I recommend Major Revisions for publication.

**Major comments**

First of all, there are lots of grammatical errors throughout the manuscript. A proofread is strongly recommended.

Second, the authors briefly describe the five parameterization schemes. Although the

description of the Voronoi scheme is sufficient, those of the other four schemes are not adequate. In particular, the authors should address/clarify the followings:

1. Fu (2007) established two ice cloud property schemes (smooth ice crystals and roughened one), both of which allow the variation of the aspect ratio. Which schemes and aspect ratio did the authors use for the present analyses (Figs. 3-9)?

2. Yi et al. (2013) included two ice cloud schemes (smooth ice crystals and roughened one) as similar to the Fu scheme. Which scheme did the authors use for the present analyses?

If the case the authors use smooth ice crystal schemes for both Fu and Yi schemes, then the authors evaluate the capabilities of the Voronoi scheme against the four schemes that are based on smoothed-surface ice crystals. This would not be a fair comparison as numerous studies have already clarified that incorporating some roughness effect into the ice cloud schemes is essential. Therefore, the authors should add one more scheme that incorporates the surface roughness (such as Yi et al. 2017ab; the MODIS C6 ice cloud scheme) for the present analyses.

These above-listed items need to be clarified before the publication of the manuscript.

**Specific comments**

1. Line 47: "Hulst, 1957" should be "van de Hulst, 1957".

2. Line 49 "Macke et al., 1996": Macke's work do not include aircraft observations, and may be irrelevant to cite this here.

3. Line 65 "Bi et al., 2013a, 2013b": Yang et al. (2013) database did not use II-TM but used the T-matrix method (Mishchenko et al., 1996).

4. Line 94: "…, and results" should be "…, and there results".

5. Line 133 "… is strong": It should be rephased to be "high".

6. Fig. 1: It would be better to show the single-scattering albedo (SSA) or the single-scattering co-albedo, instead of the scattering efficiency as it is hard to recognize the absorptivity from this particularly for weakly absorptive particles. Also, Fig. 1 indicates a second peak at the size parameter at which the transition of the computational methods between FDTD and GOIE. Is this due to a different combination of the computational methods or what occurred physically?

7. Lines 224–226 "As shown in Figure 3, …": The authors try to explain the low mass extinction coefficients at wavelengths 3.08–3.85 μm with an atmospheric window region. However, the single-scattering albedo at the corresponding wavelengths are relatively low (e.g., 0.6–0.8; Fig. 3b). Therefore, this cannot explain the low mass extinction coefficients. I suggest to check the extinction efficiency and complex refractive index of ice at corresponding wavelengths.

8. Lines. 229–231 "… than small ice particles that are closer to Rayleigh scattering": Even for small ice particles in the near-infrared band, the size parameter is much larger than the counterpart that causes the Rayleigh scattering. I suggest the authors to simply remove "that are close to Rayleigh scattering".

9. Line 242 "it is in a good agreement with the results in Zhao et al., (2019)": Too ambiguous. Please add brief descriptions in which part of results in Zhao et al. (2019) shows the agreements with your results.

10. Line 256: The downward direct flux can be different among different ice cloud parameterizations as the spectral extinction efficiency, single-scattering albedo, and asymmetry factor differ among schemes.

11. Lines 256–257 "Figure 5a1 show …": This is obvious statement and can be removed from the main text in order to let readers focus on ice cloud parameterizations.

12. Line 265 and throughout Section 4: "-10–(-40)" should be "-10 to -40". I found the same errors in several parts in Section 4, which should be corrected.

13. Line 276: "To study the ice cloud modelling capabilities" may be rephrased to be "to study the performance of ice cloud simulations with …"

14. Line 278: Please specify the 10-yr period of CERES data used for Fig. 6–7.

15. Line 281 "… are strong": This should be rephrased.

16. Lines 281–284: This statement is true if the relative fractions of liquid and ice clouds remain unchanged. Because the authors' analysis includes both liquid and ice clouds, the interpretation of the results may be mixed up. I suggest the authors to show the liquid/ice cloud fraction from both CAM5 simulations and observations.

17. Lines 294–295 and Fig. 9: The description and results are not consistent. In Fig. 9, the scheme A (Mitchell) looks the best performance. Please clarify it.

18. Fig. 7 caption: Fig. 9 should be Fig. 6.

---

## Author Comment (AC3)

Comments on "Investigation of ice cloud modeling capabilities for the irregularly shaped Voronoi models in climate simulations" by Li et al.

Anonymous Referee #3

**General comments**

This study focuses on the comparison of Voronoi model with four other ice cloud models. For the validation purpose, authors used CERES data. Authors conclude that Voronoi model-based results are closer to CERES data than results obtained from other cloud schemes. The overall goal of the study looks interesting; however, the paper is poorly organized with several mistakes in English writing, literature reference, equation citation, and so on. The discussion part is also poor.

Response: Thank you very much for your significant comments.

**Specific comments**

1. Figure 1 shows single scattering properties of only Voronoi model, though Figure 3 shows band averaged values for all cloud models. Why not to show the single scattering properties for all models in Figure 1? It can make easy to understand Figures 3 and 4 as well as other results.

Response: The main reason is that we don't have access to the database of single-scattering properties for all ice particles utilized in other four ice cloud optical property schemes.

2. It may be better to show the difference in terms of percentage (relative values) in Figure 4.

Response: According to the suggestions, we have redrawn the Figure 4 changing from absolute differences to relative percentage.

Page 31:

[Figure]

Figure 4. Relative percentage of differences of (top row) mass extinction coefficients, (second row) single scattering albedo, (third row) asymmetry factor as functions of ice particle effective diameters and shortwave bands in CAM5 between the other four schemes and Voronoi scheme.

3. There is an unclear description about particle size distribution (PSD) of ice clouds. Authors state that they utilize 14408 groups of microphysical data. Do authors use a single or multiple PSD function in this paper? For clarity, it is important to describe how PSD function is derived from 14408 groups of data to use in this study. If possible, PSD is suggested to be shown. If not possible, authors may tabulate the parameters of PSD function(s) used in this study.

Response: According to the suggestions, we have added the figure of PDSs (Figure 2 on page 28) as shown below and corresponding descriptions in section 2.2 on page 7.

[Figure]

Figure 2. Ice cloud particle size distributions based on in situ aircraft observations.

4. Figure 2 is not clear. It may be removed or improved. The methodology is well understood even without Figure 2.

Response: According to the suggestions, we have redrawn the flowchart (Figure 3) on page 28 as shown below.

[Figure]

Figure 3. Flowchart of the study

5. Equations are described in the text very randomly. For example, in section 3, Eq. 7 is described after Eq. 2. Equations and Figures are needed to appear in the text ascending order.

Response: According to the suggestions, we have reorganized the layout of equations in section 3 (page 8-10).

Page 8-9:

"To better understand the ice cloud modelling capabilities of …to the extinction coefficient in the form of Eq. (4), respectively.

$$\beta_{e,s,a} = \int_{L_{min}}^{L_{max}} \sigma_{e,s,a} n(L) dL \ , \tag{3}$$

$$\varpi = \frac{\beta_s}{\beta_e} \ , \text{or } 1 - \varpi = \frac{\beta_a}{\beta_e} \tag{4}$$

where $\sigma_{e,s}$ is the…can be defined by Eq. (5).

$$\tau = \int_z^\infty \beta_e \, dz \ , \tag{5}$$

where $z$ is … can be given by Eq. (6).

$$J(\tau; \mu; \phi) = \frac{\varpi}{4\pi} \int_0^{2\pi} \int_{-1}^1 I(\tau; \mu'; \phi') P(\mu, \phi; \mu', \phi') d\mu' d\phi'$$

$$+ \frac{\varpi}{4\pi} F_\Theta P(\mu, \phi; -\mu_0, \phi_0) e^{-\tau/\mu_0} + (1 - \varpi) B[T(\tau)] \ , \tag{6}$$

where $P$ is the phase function … expressions of ice cloud bulk optical properties as functions of $D_e$ are obtained...."

6. What is the necessity to integrate over wavelength in Eq. (7)?

Response: As you mentioned, effective diameter $D_e$ is invariant with wavelength. We have modified Eq. (7) as below,

$$D_e = \frac{3}{2} \frac{\int_{L_{min}}^{L_{max}} V(L) n(L) dL}{\int_{L_{max}}^{L_{max}} A(L) n(L) dL}$$

7. Eqs. 11-13: Equations corresponding to long wavelength bands need to be rewritten or a symbol to represent S and J may be used and stated below those equations.

Response: According to the suggestions, we have utilized a parameter $E$ in Eq. 11-13. In shortwave and longwave spectrum, $E$ is assigned by solar constants and Planck functions, respectively.

8. Authors state that the wavelength range is from 0.2 micron to 15 micron for Voronoi database and they assumed unchanged properties for wavelength larger than

15 microns. What about database for other cloud models? Do they also have such assumption? If such assumption is only for Voronoi database, what are the effects in results shown in Figure 3 and onward?

Response: As you mentioned, to ensure consistency for all schemes, we only used parameterized coefficients between 0.2 and 15 μm for the other four schemes. The wavelength range from 0.2 micron to 15 micron is sufficient for remote sensing and climate modelling applications (Yang et al., 2018; Yang et al., 2015).

9. Authors discuss about cloud forcing in Eq. (14). Can authors also discuss about the comparison of downwelling and/or upwelling fluxes for cloudy scenario between CERES and each cloud scheme? I guess comparison of fluxes rather than cloud forcing may help to better understand the performance of each model.

Response: According to the suggestions, we have added global average downwelling surface shortwave fluxes and top of atmosphere upwelling fluxes for all cases. As you mentioned, difference in net radiative components owing to different ice cloud schemes can help us to understand the performance of different schemes.

10. How water cloud is treated here is not clear. Authors may describe a more about water cloud properties and how they are merged with ice clouds in the simulation. Authors may add information (height, properties etc) related water cloud in Table 2.

Response: According to the suggestions, we have added input parameters related to liquid clouds in RRTMG simulations in Table 3, including liquid cloud top height, liquid water path, liquid effective radius.

11. What is Downward flux in Figure 5. Is it Direct+Diffuse flux? please clarify. If it is Direct+diffuse flux, why is it largely different than Difdown flux for a cloudy condition? Are clouds optically very thin for such difference?

Response: According to the suggestions, we have added more descriptions of Figure 5 (section 4.2, page 11). As you mentioned, the downward is the combination of direct and diffuse fluxes. The large difference between the downward and diffuse fluxes could be related with absence of water clouds in the RRTMG simulations.

Reference

Yang, P., Hioki, S., Saito, M., Kuo, C. P., Baum, B. A., and Liou, K. N.: A Review of Ice Cloud Optical Property Models for Passive Satellite Remote Sensing, Atmosphere, 9, 2018.

Yang, P., Liou, K. N., Bi, L., Liu, C., Yi, B. Q., and Baum, B. A.: On the Radiative Properties of Ice Clouds: Light Scattering, Remote Sensing, and Radiation Parameterization, Advances in Atmospheric Sciences, 32, 32-63, 2015.

---

## Author Comment (AC4)

Comments on "Investigation of ice cloud modeling capabilities for the irregularly shaped Voronoi models in climate simulations" by Li et al.

Anonymous Referee #4

**General comments**

The paper by Li et al. presents an analysis of a proposed broadband ice cloud scheme based on the Voronoi ice cloud particle model. The comparisons of model simulations using RRTMG and CAM5 between Voronoi and other four ice cloud schemes were carried out, indicating that the Voronoi scheme is superior to the other conventional schemes and should be sufficient for ice cloud modeling. I believe this study can be valuable to the relevant community, and it helps to better understand the ice cloud optical properties and their impact on cloud radiative effects modeling.

Overall, the study established a straightforward objective and was done in a comprehensive way. The employed scheme seemed valid and the extensive comparison was performed and discussed properly. The draw conclusions are in line with the experimental results. From my point of view, the paper is suitable for Atmospheric Chemistry and Physics, although I do have some concerns that need to be responded. To enhance the potential of the proposed scheme, I would encourage the authors to submit a revised manuscript by addressing my specific comments below:

Response: Thank you very much for your significant comments.

**Specific comments**

1. As pointed out by the other reviewers, the English language of the current manuscript requires a substantial improvement. There are a number of grammatic and wording errors (not described here as most of them have been noted by the other reviewers) in the article. A careful proofreading throughout the manuscript would be necessary.

Response: According to the suggestions, we have proofread the manuscript.

2. Please check Equation (1) at line 138 since the current layout seems weird.

Response: According to the suggestions, we have moved Eq. (1) and its corresponding descriptions to the middle of section 2 on page 6.

3. Please consider to revise Figure 2 as the flowchart does not look very helpful to me. If possible, please also include a short overview of Figure 2 in the beginning of Section 3 or reorganize this section, particularly the first paragraph. Here, you do not have to provide equation indices since you will detail them in the following subsections anyway.

Response: According to the suggestions, we have redrawn the flowchart (Figure 2) and added a brief description in the beginning of section 3 (page 8) as shown below.

Page 8: "In this study, we develop the Voronoi scheme and assess its effectiveness in comparison with Mitchell, Baum-yang05, Fu and Yi schemes. The main flowchart of this study is described in Figure 2. Five schemes are derived first and evaluated through standalone simulations in the RRTMG and multi-year simulations in the CAM5. The simulations of cloud radiative properties from different ice cloud optical property parameterizations in CAM models are measured by CERES satellite observation products. The RRTMG is utilized to understand how the different optical properties of five schemes influence the upward/downward fluxes through standalone simulations. The CAM5 is employed to evaluate the ice cloud modelling performance of the Voronoi model compared with the other four schemes in the climate system."

Flowchart (Figure 2 on page 28):

Figure 2. Flowchart of the study

4. Line 151: In Section 1, you actually only introduce the four conventional ice cloud schemes without sufficient (mathematical/technical) details. Readers would expect more details from Section 3. So, this could be another point to reorganize Section 3. Response: According to the suggestions, we have added more descriptions of the other four schemes in section 3 (page 9, 10) as shown below.

Page 9, 10: "Mitchell, Yi and Baum-yang05 schemes are developed as functions of  $D_e$  following formulation of Eq. (1-4) below. Coefficients of Mitchell scheme are obtained from ice cloud band-averaged optical properties utilized in the CAM5. Coefficients of Yi and Baum-yang05 are provided from Zhao et al. (2018). Formulation of Fu scheme is similar to Eq. (1-4) except using the generalized effective diameter (Fu, 1996) and different coefficients. Coefficients of the Fu scheme (default scheme in RRTMG) are obtained from the existing ice cloud band-averaged optical properties from RRTMG."

$$D_e = \frac{3}{2} \frac{IWC}{\rho A} , \qquad (1)$$

$$\beta = IWC(a_0 + a_1/D_e + a_2/D_e^2), \qquad (2)$$

$$1 - \omega = b_0 + b_1 D_e + b_2 D_e^2 + b_3 D_e^3 \quad , \tag{3}$$

$$g = c_0 + c_1 D_e + c_2 D_e^2 + c_3 D_e^3 , \qquad (4)$$

5. Section 3.1: I am okay with the contents. However, I would like to see a clearer structure. Each equation should normally follow the corresponding text.

Response: According to the suggestions, we have reorganized the layout of equations and corresponding illustrations in section 3 (page 8, 9) as shown below.

**Page 8, 9:**

"To better understand the ice cloud modelling capabilities of ...to the extinction coefficient in the form of Eq. (4), respectively.

$$\beta_{e,s,a} = \int_{L_{min}}^{L_{max}} \sigma_{e,s,a} n(L) dL \quad , \tag{3}$$

$$\varpi = \frac{\beta_s}{\beta_e} , \text{or } 1 - \varpi = \frac{\beta_a}{\beta_e}$$
(4)

where  $\sigma_{e,s}$  is the...can be defined by Eq. (5).

$$\tau = \int_{z}^{\infty} \beta_{e} \, dz \quad , \tag{5}$$

where z is ... can be given by Eq. (6).

$$J(\tau;\mu;\phi) = \frac{\varpi}{4\pi} \int_{0}^{2\pi} \int_{-1}^{1} I(\tau;\mu';\phi') P(\mu,\phi;\mu',\phi') d\mu' d\phi' + \frac{\varpi}{4\pi} F_{\theta} P(\mu,\phi;-\mu_{0},\phi_{0}) e^{-\tau/\mu_{0}} + (1-\varpi) B[T(\tau)], \qquad (6)$$

where *P* is the phase function ...."

6. Line 269: It sounds unclear to me based on what quality criteria the authors ranked the five models.

Response: Five schemes were sorted from large to small values of upward/downward fluxes.

7. Lines 293-296: Please explain Figure 9 in detail, more explicitly, why the Voronoi model performed the best. So far, I am not convinced by the statement in the current manuscript "...differences box of Voronoi scheme are most concentrated on the zero ...".

Response: According to the suggestions, the box plot is to describe the data of five statistic: the minimum, first quartile, median, and the third quartile and the maximum value. The closer the median line is to the zero line, the more evenly the boxes are

**distributed on both sides of the zero line, the better the scheme is.**

8. An additional appendix including all acronyms and abbreviations used in the manuscript would be useful to readers.

Response: According to the suggestions, we have added a table (Table 1) contains all acronyms in the manuscript.

**Page 21, 22:**

[revised manuscript text omitted]

---

## Author Response (AR1)

Comments on "Investigation of ice cloud modeling capabilities for the irregularly shaped Voronoi models in climate simulations" by Li et al.

Anonymous Referee #1

**General comments**

The paper assesses the performance of the Voronoi ice crystal model on the broadband radiative transfer simulations as well as climate simulations with CAM5 through the comparisons to other four ice cloud parameterizations including Mitchell, Fu, Baum-Yang, and Yi schemes. The Voronoi scheme exhibits relatively lower asymmetry factor and higher single-scattering albedo in the visible to near-infrared wavelength domain, than the other four schemes, resulting in more reflective ice clouds in shortwave radiative transfer simulations. The comparisons of the net cloud radiative effects between CERES observations and 10-yr CAM5 simulations among these ice cloud parameterization schemes suggest that the Voronoi scheme outperforms the other four parameterization schemes. The authors conclude that the Voronoi scheme can minimize the differences of the global TOA SWCF between the satellite-based measurements and the CAM5 simulation counterparts compared to other four schemes. This paper sufficiently describes the background and introduction, and methods. However, the result section contains inadequate discussions in the interpretation of the results. In particular, the authors should add more detailed descriptions on the five parameterization schemes. Also, there are numerous grammatical/language errors, and several sentences need to be rephrased. The topic presented in this study is suitable for Atmospheric Chemistry and Physics, and therefore I recommend Major Revisions for publication.

Response: Thank you very much for your significant comments.

**Major comments**

1. First of all, there are lots of grammatical errors throughout the manuscript. A proofread is strongly recommended.

Response: According to the suggestions, we have proofread the manuscript.

2. Second, the authors briefly describe the five parameterization schemes. Although the description of the Voronoi scheme is sufficient, those of the other four schemes are not adequate. In particular, the authors should address/clarify the followings:

- Fu (2007) established two ice cloud property schemes (smooth ice crystals and roughened one), both of which allow the variation of the aspect ratio. Which schemes and aspect ratio did the authors use for the present analyses (Figs. 3-9)?
- 2. Yi et al. (2013) included two ice cloud schemes (smooth ice crystals and roughened one) as similar to the Fu scheme. Which scheme did the authors use for the present analyses?

If the case the authors use smooth ice crystal schemes for both Fu and Yi schemes, then the authors evaluate the capabilities of the Voronoi scheme against the four schemes that are based on smoothed-surface ice crystals. This would not be a fair comparison as numerous studies have already clarified that incorporating some roughness effect into the ice cloud schemes is essential. Therefore, the authors should add one more scheme that incorporates the surface roughness (such as Yi et al. 2017ab; the MODIS C6 ice cloud scheme) for the present analyses.

These above-listed items need to be clarified before the publication of the manuscript. Response: According to the suggestions, we have added more illustrations about the other four schemes in section 1 paragraph 3 (page 4). Modified descriptions of Fu and Yi schemes are as follows: "Fu (1996) derived an ice cloud optical parameterization (referred to as Fu scheme hereafter), in which optical properties have been parameterized as functions of ice water content and generalized effective size based on the randomly oriented hexagonal ice particle. ...Yi et al. (2013) developed a parameterization (referred to as Yi scheme hereafter) based on a general habit mixture model that includes nine pristine habits with severely surface roughness."

In the study, we have accounted for the effects of roughened ice particles in scheme of Yi et al. (2013).

**Specific comments**

1. Line 47: "Hulst, 1957" should be "van de Hulst, 1957".

2. Line 49 "Macke et al., 1996": Macke's work do not include aircraft observations, and may be irrelevant to cite this here.

3. Line 65 "Bi et al., 2013a, 2013b": Yang et al. (2013) database did not use II-TM but used the T-matrix method (Mishchenko et al., 1996).

Response: According to the suggestions, we have corrected the wrong citations.

4. Line 94: "..., and results" should be "..., and there results".

5. Line 133 "... is strong": It should be rephased to be "high".

Response: According to the suggestions, we have modified the irrational expressions.

6. Fig. 1: It would be better to show the single-scattering albedo (SSA) or the single-scattering co-albedo, instead of the scattering efficiency as it is hard to recognize the absorptivity from this particularly for weakly absorptive particles. Also, Fig. 1 indicates a second peak at the size parameter at which the transition of the computational methods between FDTD and GOIE. Is this due to a different combination of the computational methods or what occurred physically?

Response: According to the suggestions, we have changed the scattering efficiency to the single-scattering albedo (SSA) in Fig. 1. And the main reason is that the second peak is caused by transitions of different computational methods in visible and near-infrared wavelength.

7. Lines 224 – 226 "As shown in Figure 3, …": The authors try to explain the low mass extinction coefficients at wavelengths  $3.08 - 3.85 \mu m$  with an atmospheric window region. However, the single-scattering albedo at the corresponding wavelengths are relatively low (e.g., 0.6 - 0.8; Fig. 3b). Therefore, this cannot explain the low mass extinction coefficients. I suggest to check the extinction efficiency and complex refractive index of ice at corresponding wavelengths.

Response: According to the suggestions, we have checked the refractive index of ice shown in Figure 1 below, and found it could because that the real part of the refractive index sharply decreases near 3  $\mu$ m and reach the minimum at 3  $\mu$ m (Warren and Brandt, 2008; Yang et al., 2013). This could result in a minimum value of mass

extinction coefficients.

Figure 1. (b) Real part of the refractive index;

(c) Imaginary part of the refractive index, cited from Yang et al. (2013).

8. Lines. 229 – 231 "… than small ice particles that are closer to Rayleigh scattering": Even for small ice particles in the near-infrared band, the size parameter is much larger than the counterpart that causes the Rayleigh scattering. I suggest the authors to simply remove "that are close to Rayleigh scattering".

Response: According to the suggestions, we have removed the irrational expressions.

9. Line 242 "it is in a good agreement with the results in Zhao et al., (2019)": Too ambiguous. Please add brief descriptions in which part of results in Zhao et al. (2019) shows the agreements with your results.

Response: According to the suggestions, we have added descriptions as follows: "This highest asymmetry factor of the Mitchell scheme is also found when comparing with other schemes in the study of Zhao et al. (2018)" in section 4.1 paragraph 2 (page 14).

10. Line 256: The downward direct flux can be different among different ice cloud parameterizations as the spectral extinction efficiency, single-scattering albedo, and asymmetry factor differ among schemes.

Response: According to the suggestions, we have modified the inappropriate expressions in section 4.2 (page 13).

11. Lines 256 - 257 "Figure 5a1 show ...": This is obvious statement and can be

removed from the main text in order to let readers focus on ice cloud parameterizations.

Response: According to the suggestions, we have removed the irrational expressions.

12. Line 265 and throughout Section 4: "-10 - (-40)" should be "-10 to -40". I found the same errors in several parts in Section 4, which should be corrected.

13. Line 276: "To study the ice cloud modelling capabilities" may be rephrased to be "to study the performance of ice cloud simulations with …"

Response: According to the suggestions, we have corrected the irrational expressions.

14. Line 278: Please specify the 10-yr period of CERES data used for Fig. 6 - 7.

Response: According to the suggestions, we have added descriptions in section 2.3 paragraph 2 (7). The temporal period of CERES products utilized in this study is from 2001 to 2010.

15. Line 281 "... are strong": This should be rephrased.

Response: According to the suggestions, we have corrected the irrational expressions.

16. Lines 281 – 284: This statement is true if the relative fractions of liquid and ice clouds remain unchanged. Because the authors'analysis includes both liquid and ice clouds, the interpretation of the results may be mixed up. I suggest the authors to show the liquid/ice cloud fraction from both CAM5 simulations and observations.

Response: Total cloud radiative effects are shown because the radiation scheme in CAM5 unable to treat liquid and ice clouds individually, thus we cannot separate the effects of ice clouds from the total amounts. The modifications of total cloud radiative effects can only be attributed to the difference of different ice cloud scheme adopted in radiation scheme in CAM5 with unchanged liquid cloud scheme.

17. Lines 294 - 295 and Fig. 9: The description and results are not consistent. In Fig.9, the scheme A (Mitchell) looks the best performance. Please clarify it.

Response: Given the Mitchell scheme overestimates in the tropics and underestimates in the middle to high latitudes in both hemispheres, the positive and negative differences can produce compensating biases, which result in that the difference of globally averaged SWCF and LWCF of Mitchell scheme is closest to the zero line.

18. Fig. 7 caption: Fig. 9 should be Fig. 6.

Response: We have corrected this error in Figure 7 caption.

**Reference**

- Fu, Q. A.: An accurate parameterization of the solar radiative properties of cirrus clouds for climate models, J Climate, 9, 2058-2082, 1996.
- Warren, S. G. and Brandt, R. E.: Optical constants of ice from the ultraviolet to the microwave: A revised compilation, J Geophys Res-Atmos, 113, 2008.
- Yang, P., Bi, L., Baum, B. A., Liou, K. N., Kattawar, G. W., Mishchenko, M. I., and Cole, B.: Spectrally Consistent Scattering, Absorption, and Polarization Properties of Atmospheric Ice Crystals at Wavelengths from 0.2 to 100 μm, J Atmos Sci, 70, 2013.
- Yi, B. Q., Yang, P., Baum, B. A., L'Ecuyer, T., Oreopoulos, L., Mlawer, E. J., Heymsfield, A. J., and Liou, K. N.: Influence of Ice Particle Surface Roughening on the Global Cloud Radiative Effect, J Atmos Sci, 70, 2794-2807, 2013.
- Zhao, W. J., Peng, Y. R., Wang, B., Yi, B. Q., Lin, Y. L., and Li, J. N.: Comparison of three ice cloud optical schemes in climate simulations with community atmospheric model version 5, Atmos Res, 204, 37-53, 2018.

Comments on "Investigation of ice cloud modeling capabilities for the irregularly shaped Voronoi models in climate simulations" by Li et al.

Anonymous Referee #2

**General comments**

This paper addressed the important problem about the modeling capability of ice cloud radiative forcing in climate simulations. An irregularly shaped Voronoi ice cloud particle model which was proven to be effective and efficient in satellite remote sensing retrieval purposes has been implemented in the RRTMG RTM and CAM5 climate model. Comparisons of modeling results with the Voronoi model along with the other four previously proposed ice cloud models are carried out. Further comparison between model results and the CERES SYN1deg radiative fluxes indicates that the Voronoi model provides the closest cloud radiative forcing to observation. This study could be a good supplement to understand the influence of ice cloud optical properties on simulated cloud radiative effects. The topic of this paper is within the scope of the journal of Atmospheric Chemistry and Physics. But unfortunately, the paper is not acceptable in the present form due to various issues.

Response: Thank you very much for your significant comments.

**Major comments**

1. Overall, the quality of this paper does not meet the standard of ACP. The problem is all-round, from simple wording and sentence expressions to the quality of figures and tables, data analysis, conclusions, and so on. There are so many things to be improved to make the paper better (please see details below).

Response: Thank you very much for your significant comments. We have revised the manuscript accordingly.

2. The authors should be cautious about the definition and the use of abbreviations. Several abbreviations are defined again and again. While some other abbreviations are defined and never used again. Abbreviations like AGCM are used without definition. This may be just subtle issue but it could be an indicator of how the paper is carelessly prepared ...

Response: According to the suggestions, we have removed all irrational abbreviations.

3. The authors need to pay more attention to the way to cite papers. Some of the names of the authors are wrong! For example, Line 47, "Hulst" should be "van de Hulst"; Line 68, "Labonnote" should be "C.-Labonnote". Incorrect citation formats also exist, for example, at Line 61, 63, 83, 86, etc. This is another indicator that the paper undergoes insufficient examination before submission.

Response: According to the suggestions, we have corrected the citations throughout the manuscript.

4. It looks like the "Baum-Yang" scheme in this paper is different from the "Baum-Yang" scheme in Wang et al. (2018). It may be better to rename the schemes to avoid confusion when the readers are comparing the two studies.

Response: As you mentioned, these two schemes are different from each other. We have renamed the ice cloud parameterization scheme formed by Baum et al. (2005b) as "Baum-yang05 scheme" hereafter in the manuscript.

5. Among the various schemes, Fu scheme actually has different definitions of effective diameter (see Fu et al., 1997). So the question is how can the Fu scheme be compared directly with the other schemes?

Response: As you mentioned, the Fu scheme uses the generalized parameter  $D_{ge}$  (as shown below Eq. (1)), the other four schemes use the effective parameter  $D_e$  (as shown below Eq. (2)). As  $D_{ge}$  can be converted to  $D_e$  by a constant,  $D_{ge}$  is unified to  $D_e$  for consistency.

$$D_{e} = \frac{3}{2} * \frac{IWC}{\rho A},\tag{1}$$

$$D_{ge} = \frac{2\sqrt{3}}{3} * \frac{IWC}{\rho A},$$
(2)

6. These Line 89-92: What's the point of mentioning CIESM at this point? Since CIESM is no different from CESM regarding the ice cloud scattering properties, there seems no need to mention it at all. After all, the authors are actually using the original CAM5, isn't it?

Response: Yes. As you pointed out, we actually use the CAM5 model in this study. According to the suggestions, we have removed the Line 89-92 on Page 2 related to CIESM.

7. I don't like the way the authors organized the figure panels. It's strange to me to use panel a1, a2, ... and b1, b2, ... in a same figure. Please consider following the conventional panel naming habit of (a), (b), (c), ...

Response: According to the suggestions, we have renamed all figure panels using (a), (b), (c).

8. I don't like the organization of section 3 either. Particularly, Line 140-159 is a mess. It may not be a good idea to briefly referring to something you will mentioned in detail later. It makes no sense and just add to the confusion of the reader.

Response: According to the suggestions, we have rewritten Line 140-159 in section 3 as shown below.

Page 7, 8: "In this study, we develop the Voronoi scheme and assess its effectiveness in comparison with Mitchell, Baum-yang05, Fu and Yi schemes. The main flowchart of this study is described in Figure 2. Five schemes are derived first and evaluated through standalone simulations in the RRTMG and multi-year simulations in the CAM5. The simulations of cloud radiative properties from different ice cloud optical property parameterizations in CAM models are evaluated by CERES satellite observation products. The RRTMG is utilized to understand how the different optical properties of five schemes influence the upward/downward fluxes through standalone simulations. The CAM5 is employed to evaluate the ice cloud modelling performance of the Voronoi model compared with the other four schemes in the climate system."

9. More details about the particle size distributions should be given. The authors may add a figure to show how the PSD looks like.

Response: According to the suggestions, we have added the figure of PSD (Figure 2) in the manuscript as shown below.

Page 29:

Figure 2. Ice cloud particle size distributions for different temperature.

10. Why do the authors choose to use CERES\_SYN1deg\_Ed4A products for comparison with GCM modeling results? What is the temporal range of data used? Usually, the CERES EBAF dataset is a better choice for this purpose. Since the authors' choice will apparently affect the evaluation results, it is quite necessary for the authors to elaborate the reasons more convincingly.

Response: According to the suggestions, we have used CERES EBAF Ed4.1 products from January 2001 to December 2010 for validation in this study.

11. The authors should do a better job of relating the optical properties of different ice cloud models with the simulated SWCF and LWCF. It is still confusing to me that why the Voronoi model possesses the lowest asymmetry factor in the SW but however exhibits the lowest SWCF compared with the other models? In short, why the Voronoi model could be the better choice?

Response: Firstly, the difference of asymmetry factor of different schemes is mainly attributed to different ice particle habits or shapes utilized in each scheme. Secondly, as shown in the Figure 1 below, the difference of ice cloud bulk optical properties between different schemes indicate that the Voronoi scheme possesses the lowest asymmetry factor compared to the other four schemes. The lowest asymmetry factor of Voronoi scheme can result in more reflected TOA radiation than the other four schemes. The difference of reflected radiation for cloudy and clear conditions are reduced, which can result in closer SWCF of the Voronoi scheme to the satellite observations.

Figure 4. The (a) Mitchell, (b) Fu, (c) Baum-yang05 and (d) Yi schemes minus the Voronoi scheme differences (%) in (top row) mass extinction coefficients, (middle row) single-scattering albedo and (bottom row) asymmetry factor as functions of effective diameter and 14 shortwave bands.

12. Many grammar mistakes and sentence errors could be found in the manuscript. The authors should pay more attention to polish the English language. Several captions of figures and tables also need to be rephrased. For example, Table 3 and Figure 9 all miss the units. While caption of Figure 5 is too complicated to understand.

Response: According to the suggestions, we have proofread the manuscript, added units to the Table 3 and Figure 9 in the manuscript and rewritten the caption of Figure 5.

**Minor comments**

1. Line 17-18: "While abundant irregularly shaped ice particle habits present a challenge for modelling ice clouds." – please be clear about what challenge is the authors referring to.

Response: We have modified the sentence as "Complexities of ice particle habits/shapes and sizes make it difficult to select a representative ice scattering model for simulating the real ice cloud scattering properties.".

2. Line 24: There may be no need to express the names of the other four schemes since they are not used again in the abstract, and the readers still could not understand what the names are referring to.

Response: According to the suggestions, we have removed the names of the other four schemes in the abstract.

3. Line 25, 27, 30: RRTMG, CERES, SW and LW are never used again in the abstract. May not need to define the abbreviations.

Response: According to the suggestions, we have deleted the unused abbreviation names in the abstract.

4. Line 91: "CAM5 in CIESM was modified with several new schemes"- what are the changes which relate to this study?

Response: Changes include cloud macrophysics including cloud fraction and condensation using PDF cloud scheme (Qin et al., 2018) and cloud microphysics using single-ice scheme from Zhao et al. (2017). The ice cloud parameterization scheme (Mitchell scheme) remains unchanged in the CAM5 in CIESM. However, according to the aforementioned suggestions, we have removed contents related to CIESM in the manuscript.

5. Line 101-105: Very complicated sentence which contains error.

Response: According to the suggestions, we have rewritten these expressions in the manuscript as shown below.

**Page 5:**

"Ishimoto et al. (2012) developed an irregularly shaped Voronoi model based on in situ microphysical measurements. Letu et al. (2016) compared the Voronoi model with the conventional general habitat mixture (GHM) (Baum et al., 2011), IHMs (C.-Labonnote et al., 2001), 5-plate aggregate (Baum et al., 2005a, 2011), and the ensemble ice particle model (Baran and Labonnote, 2007) through minimizing the difference between the observed polarized reflectivity and the simulations. The results indicated that the irregularly shaped Voronoi model outperformed with the measured polarized reflectivity from the POLDER observations."

6. Line 109: "has proven" should be "has been proven"

Response: According to the suggestions, we have corrected this irrational expression on page 5.

7. Line 138: It's very odd to see equation (1) here without any explanation.

Response: According to the suggestions, we have reorganized the layout of Eq. (1) as shown below.

Page 6:

"The definition of SZP is shown below,

$$SZP = \frac{\pi L}{\lambda} , \qquad (1)$$

where *L* is the ice particle maximum diameter."

8. Line 148: What is the temperature used in the Planck function?

Response: We defined the Planck function assuming a cloud temperature of 233K according to Liou (1992).

9. Line 157: "to validate the cloud radiative properties" – do you mean "cloud radiative Forcing"?

Response: Yes, we have replaced "cloud radiative properties" with "cloud radiative forcing" on page 7.

10. Line 166-167: Sentence error.

Response: We have corrected the Line 166-167 as shown below.

Page 8:

" $\sigma_{e,s}$  is the extinction and scattering cross section, respectively (See Table 1 for a list of acronyms), and  $\sigma_a$  is the absorption cross section given by  $\sigma_a = \sigma_e - \sigma_s$ ."

11. Line 197-198: What is "standard tropics"? How are the 60 vertical levels distributed? Please give a reference.

Response: The term "standard tropics" means a U.S. Standard reference atmospheric model profile consisting of vertical profile for temperature and gas mixing ratios designed for tropic cases (15N annual average) (Anderson et al., 1986). The 60

vertical profiles are distributed from bottom pressure 1013.0 mb to the top pressure 0.0003 mb.

12. Line 201: "the same with" should be "the same as" Response: Corrected.

Page 11:

"LWCF is defined the same as SWCF but for LW spectrum."

13. Line 242: should be Zhao et al. (2018)?Response: Agreed, we have corrected the citation.

Page 12:

"This highest asymmetry factor of the Mitchell scheme is also found when comparing among other schemes in the study of Zhao et al. (2018)."

14. Line 264-266: Please change a way to express the range of values since the present form easily cause confusion.

Response: According to the suggestion, we have rewritten the sentence as "Figure 5 shows 6-30 W/m2 differences in TOA upward fluxes, 10-40 W/m2 differences in surface downward diffuse flux, 10-30 W/m2 differences in surface net fluxes, and 8-42 W/m2 differences in TOA net fluxes owing to five different ice cloud schemes.".

15. What is the version of RRTMG used?

Response: The version of the RRTMG used in our study is the current radiative code applied in the CAM5 (Mlawer et al., 1997; Iacono et al., 2008; available from <a href="http://rtweb.aer.com">http://rtweb.aer.com</a>).

16. Line 320: The authors may need to specify the contribution of all authors.

Response: According to the suggestions, we have added more specific contributions of each author as shown below.

Page 16: "Ming Li developed the ice cloud optical property parameterizations (Voronoi scheme) based on the single-scattering properties of Voronoi models, compared the band-averaged optical properties of the Voronoi scheme with the other four schemes (Mitchell, Yi, Baun-yang05 and Fu). Ming Li also compared the

upward/downward flux profiles from five schemes through RRTMG standalone simulations and radiative properties of five schemes in CAM5 model simulations, as well as downloaded the CERES products and wrote the initial draft of this manuscript. Husi Letu designed the aims and structures of this study and assisted in developing the parameterization of ice cloud optical properties based on the Voronoi models. Husi Letu also provided the single-scattering property database of Voronoi models and helped in analyzing the single-scattering properties of Voronoi models, as well as guided the writings and revisions of the manuscript. Yiran Peng and Yanluan Lin assisted in developing the ice cloud optical property parameterization of the Voronoi scheme and provided the climate models, as well as guided the settings of climate model runs and reviewing the manuscript. Hiroshi Ishimoto developed the single-scattering property database of Voronoi models, provided the database of Voronoi models and helped in the parameterization of ice cloud optical properties based on the single-scattering properties of Voronoi models. Takashi Y. Nakajima provided the single-scattering property database of Voronoi models, especially assisted in guiding the flowchart of this study and reviewed the manuscript. Anthony Baran guided the developm

---

## Referee Report (RR1)

Manuscript number: acp-2021-208

Full title: Investigation of ice cloud modelling capabilities for the irregularly shaped Voronoi ice scattering models in climate simulations

Author(s): Li et al.

I appreciate that the authors have provided reasonable responses to most of my comments. Please find follow-up comments below to improve the manuscript. However, there are still numerous errors throughout the manuscript. I even doubt if the authors did proofread it in response to my previous comment! In particular, a number of grammatical errors, inconsistent Table numbers, and inconsistent captions are found, which should be corrected. The topic presented in this study is suitable for Atmospheric Chemistry and Physics, and a minor revision is required for publication.

**Specific comments**

1. Response to Comment #7: As the bulk mass extinction efficiency is a function of the bulk extinction efficiency, bulk geometric particle projected area and volume, and ice density. Since the geometric parameters do not change over spectral wavelengths, the bulk extinction efficiency should have a local minimum at the corresponding wavelength domain. The explanation that the authors have provided here makes physically no sense to me. If the minimum value of the real part of the ice refractive index leads to a minimum mass extinction coefficient, please provide the physical reason why it does.

2. Response to Comment #16: If CAM5 unable to treat liquid and ice clouds individually, the present analysis may involve potential uncertainty associated with the ice/liquid fraction, which should be mentioned in the manuscript.

3. Line 139 "The single-scattering albedo at both wavelengths is close to 1, which is

related to the high values of the imaginary part in the refractive index.": High value of the imaginary part of the refractive index indicates very absorptive, and therefore SSA should be low.

4. Line 159: "compared" should be "compare".

5. Line 228 "see Table 1": This should be Table 2, shouldn't be?

6. Line 240 "The CIESM is run in two ways:" it has two verbs. It seems to me that a proofread was inadequate. Please double check the grammar in the manuscript.

7. Captions in Figs. 4-5: Although the figures show spectral bulk optical properties between 0.2-15 μm, the captions indicate the 14 shortwave bands, which is inconsistent.

8. Lines 269–271: Why the Voronoi model has larger mass extinction coefficients than Fu, Yi, and Baum-yang05 model counterparts?

---

## Author Response (AR3)

Comments on "Investigation of ice cloud modelling capabilities for the irregularly shaped Voronoi ice scattering models in climate simulations" by Li et al.

Anonymous Referee #1

**General comments**

I appreciate that the authors have provided reasonable responses to most of my comments. Please find follow-up comments below to improve the manuscript. However, there are still numerous errors throughout the manuscript. I even doubt if the authors did proofread it in response to my previous comment! In particular, a number of grammatical errors, inconsistent Table numbers, and inconsistent captions are found, which should be corrected. The topic presented in this study is suitable for Atmospheric Chemistry and Physics, and a minor revision is required for publication.

Response: Thank you very much for your significant comments.

**Specific comments**

1. Response to Comment #7: As the bulk mass extinction efficiency is a function of the bulk extinction efficiency, bulk geometric particle projected area and volume, and ice density. Since the geometric parameters do not change over spectral wavelengths, the bulk extinction efficiency should have a local minimum at the corresponding wavelength domain. The explanation that the authors have provided here makes physically no sense to me. If the minimum value of the real part of the ice refractive index leads to a minimum mass extinction coefficient, please provide the physical reason why it does.

Response: As you mentioned, we have analyzed the spectral features of extinction efficiency $Q_{ext}$ of Voronoi ice particles. We choose six Voronoi ice particles with different effective diameters $D_e$ and show their spectral $Q_{ext}$ in Figure 1. As shown in Figure 1, we find that the $Q_{ext}$ for all particles have a minimum near the $\lambda$ equals

3 μm, which can lead to the minimum mass extinction coefficient $K_{ext}(\lambda)$ at the corresponding $\lambda$ region according to Eq. (1).

$$K_{ext}(\lambda) = \frac{\int_{L_{min}}^{L_{max}} Q_{ext}(\lambda, L) A(L) n(L) dL}{\rho_{ice} \int_{L_{min}}^{L_{max}} V(L) n(L) dL}$$

(1)

[Figure]

Figure 1. The spectral features of Voronoi ice particle $Q_{ext}(\lambda)$ for six particles with different $D_e$.

2. Response to Comment #16: If CAM5 unable to treat liquid and ice clouds individually, the present analysis may involve potential uncertainty associated with the ice/liquid fraction, which should be mentioned in the manuscript.

Response: According to the suggestions, we have added descriptions of potential uncertainty associated with the ice/liquid cloud fraction in section 4.3 as shown below.

Line 317-319 on Page 14: "Results show that the SWCF of CIESM simulations are stronger than the EBAF results in tropical regions. This might be because that the CIESM overestimates ice/liquid cloud fraction in low-latitudes compared with the observations (Eidhammer et al., 2014; Kay et al., 2016)."

3. Line 139 "The single-scattering albedo at both wavelengths is close to 1, which is related to the high values of the imaginary part in the refractive index.": High value of the imaginary part of the refractive index indicates very absorptive, and therefore SSA should be low.

Response: According to the suggestions, we have corrected this error as shown below.

Line 141 on Page 6: "The single-scattering albedo at both wavelengths is close to 1, which is related to the high values of the real part in the refractive index."

4. Line 159: "compared" should be "compare".

Response: As you mentioned, we have corrected the error in Line 161 on Page 7.

5. Line 228 "see Table 1": This should be Table 2, shouldn't be?

Response: Yes. As you mentioned, we have modified the error in Line 230 on Page 11.

6. Line 240 "The CIESM is run in two ways:" it has two verbs. It seems to me that a proofread was inadequate. Please double check the grammar in the manuscript.

Response: According to the suggestions, we have checked the manuscript, and we have rewritten the above sentence as shown below.

Line 242 on Page 11: "The CIESM is operated in two ways"

7. Captions in Figs. 4-5: Although the figures show spectral bulk optical properties between 0.2-15 μm, the captions indicate the 14 shortwave bands, which is inconsistent.

Response: As you mentioned, we have corrected captions in Figure 4 and 5.

8. Lines 269 – 271: Why the Voronoi model has larger mass extinction coefficients than Fu, Yi, and Baum-yang05 model counterparts?

Response: According to the above Eq. (1), this is related with the extinction efficiency, projected area, and volume for ice particles. We don't have access to the single-scattering properties of ice particle habits utilized in the Fu, Yi, and Baum-yang05 schemes. So, it is difficult for us to quantify the differences in extinction properties among different ice particle habits.

Further, the Figure 6 in the manuscript displays that the Voronoi model has lower downward fluxes than Fu, Yi, and Baum-yang05 schemes. The extinction coefficient determines the cloud optical thickness and directly reduces the downward fluxes. The larger the extinction coefficient, the lower the downward fluxes. Thus, the results in Figure 6 are consistent with the conclusion that the Voronoi model has larger mass extinction coefficients than Fu, Yi, and Baum-yang05 schemes.

Reference:

Eidhammer, T., Morrison, H., Bansemer, A., Gettelman, A., and Heymsfield, A.: Comparison of ice particle characteristics simulated by the Community Atmosphere Model (CAM5) with in-situ observations, Atmospheric Chemistry and Physics Discussions, 14, 10.5194/acpd-14-7637-2014, 2014.

Kay, J., Bourdages, L., Miller, N., Morrison, A., Yettella, V., Chepfer, H., and Eaton, B.: Evaluating and improving cloud phase in the Community Atmosphere Model version 5 using spaceborne lidar observations, Journal of Geophysical Research: Atmospheres, 121, 10.1002/2015JD024699, 2016.